# Thirst interneurons that promote water seeking and limit feeding behavior in *Drosophila*

**Dan Landayan[1†], Brian P Wang[1], Jennifer Zhou[2], Fred W Wolf[1,2]\***

[1]Quantitative and Systems Biology Graduate Program, UC, Merced, United States;
[2]Department of Molecular and Cell Biology, UC, Merced, United States

**Abstract** Thirst is a motivational state that drives behaviors to obtain water for fluid homeostasis. We identified two types of central brain interneurons that regulate thirsty water seeking in *Drosophila*, that we term the Janu neurons. Janu-GABA, a local interneuron in the subesophageal zone, is activated by water deprivation and is specific to thirsty seeking. Janu-AstA projects from the subesophageal zone to the superior medial protocerebrum, a higher order processing area. Janu-AstA signals with the neuropeptide Allatostatin A to promote water seeking and to inhibit feeding behavior. NPF (*Drosophila* NPY) neurons are postsynaptic to Janu-AstA for water seeking and feeding through the AstA-R2 galanin-like receptor. NPF neurons use NPF to regulate thirst and hunger behaviors. Flies choose Janu neuron activation, suggesting that thirsty seeking up a humidity gradient is rewarding. These findings identify novel central brain circuit elements that coordinate internal state drives to selectively control motivated seeking behavior.

**\*For correspondence:**
fwolf@ucmerced.edu

**Present address:** [†]Department of Psychiatry and Behavioral Sciences, Stanford University, Stanford, United States

**Competing interests:** The authors declare that no competing interests exist.

## Introduction

Thirst is the neural representation of the internal state caused by dehydration or decreased blood volume. Thirst motivates a coordinated series of water seeking and intake behaviors across diverse animal phyla. Terrestrial animals with unpredictable access to patchily distributed sources of water, such as the fly *Drosophila melanogaster*, have evolved robust brain circuitry to rapidly detect water loss to drive goal-directed seeking and ingestion behaviors.

In flies, water ingestion and humidity preference studies led to the discovery of sensory and central brain circuit elements that control behavioral and hormonal aspects of thirst. Flies rapidly detect humidity gradients via humid-sensing (Ir68a receptor) and dry-sensing (Ir40a receptor) hygrosensory neurons that are housed in the sacculus on the antennae (*Enjin et al., 2016*; *Ji and Zhu, 2015*; *Knecht et al., 2016*; *Knecht et al., 2017*). The hygrosensory neurons project to dedicated glomeruli in the antennal lobe, where they synapse onto second order projection neurons to send information to the mushroom bodies, lateral horn, and broadly to other areas of the protocerebrum (*Frank et al., 2017*; *Marin et al., 2020*). Water taste is facilitated by water-sensing (PPK28 mechano-receptor) gustatory neurons; taste is critical for water consumption in thirsty flies (*Cameron et al., 2010*; *Lau et al., 2017*). A pair of interoceptive SEZ neurons (ISN) that detect osmolarity are located in the suboesophageal ganglion zone (SEZ); they promote water consumption and reciprocally inhibit food intake when inhibited by high interstitial fluid osmolarity (*Jourjine et al., 2016*). The ISNs are regulated by the mechanoreceptor Nanchung that responds to changes in hemolymph osmolarity to signal water balance status, and by adipokinetic hormone that signals aspects of energy store status to the brain.

These sensory and interoceptive inputs regulate central brain circuitry to facilitate ingestive behavior and initiate the state of thirst. How the thirsty state is represented in the brain remains to be characterized. Functional mapping of water reward memories led to the discovery of leucokinin

peptidergic neurons, whose activity is regulated by thirst, and that regulate memories of thirst through dopaminergic connections into the mushroom body learning and memory circuitry (*Ji and Zhu, 2015*; *Lin et al., 2014*; *Senapati et al., 2019*). Thirsty attraction to humidity may also route through some of the same dopamine neurons (*Lin et al., 2014*). Hunger, which drives a pattern of ingestive behaviors similar to that of thirst, is also conveyed through dopamine neurons and the mushroom body circuitry (*Tsao et al., 2018*). However, the role of hunger circuitry in water seeking is not yet known. Finally, the pars intercerebralis, a neuroendocrine brain region, contains ion transport peptide (ITP) hormone expressing neurons (*Gáliková et al., 2018*; *Nässel and Vanden Broeck, 2016*). Ion transport peptide bidirectionally controls thirst and hunger behaviors, and it regulates water balance in the body as a hormone.

Understanding how fundamental motivational states are related at the circuit level will help uncover the hierarchy of behavioral choice and its underlying neural mechanisms. Thirst in mammals has complex hierarchical interactions with other behavioral states, including hunger and sleep (*Sutton and Krashes, 2020*). Thirst and hunger are particularly tightly interconnected. For example, thirst can override hunger drive (dehydration-anorexia), and feeding induces thirst (prandial thirst) (*Watts and Boyle, 2010*). Manipulation of either cortical or hindbrain neurons in mice can simultaneously impact both hunger and thirst driven ingestive behavior (*Eiselt et al., 2021*; *Gong et al., 2020*). Moreover, thirst-driven water anticipation and subsequent intake changes the activity of a remarkably broad set of neurons throughout the mouse brain, in a manner that is time-locked to specific behavioral steps (*Allen et al., 2019*). This reveals the complexity of even an apparently simple motivational state, and it predicts interactions with other brain functions. Flies, exhibiting hunger:thirst and hunger:sleep interdependencies with apparently equivalent ethological relationships, will be useful for elaborating circuit motifs for behavioral state interactions, particularly at the level of individual neurons (*Melnattur and Shaw, 2019*).

To identify central brain neurons critical for thirst, we used an open-field behavioral assay that allows neural manipulation of freely moving animals as they evaluate the presentation of a point source of water or food. An unbiased neuroanatomical activation screen identified two bilateral pairs of neurons, Janu-GABA and Janu-AstA, whose activity is necessary and sufficient for promoting thirsty water seeking, but that have limited effect on fluid consumption. Flies choose to continuously activate the Janu neurons, suggesting that water seeking up a humidity gradient is rewarding. The Janu-GABA local interneurons are specific to thirsty water seeking. In contrast, the Janu-AstA neurons, that release the neuropeptide Allatostatin A (AstA), also inhibit feeding behavior. Neuropeptide F (NPF) neurons are a postsynaptic target of Janu-AstA neurons through the AstA-R2 receptor in the superior medial protocerebrum. NPF reciprocally regulates water seeking and feeding behavior. Thus, we uncovered central brain interneurons that encode water seeking and that simultaneously suppress food seeking, indicating that thirst and hunger are coordinated homeostatic states while animals are actively seeking.

## Results

### Anatomical activation screen identifies water-seeking neurons

Thirsty flies are motivated to seek a palatable source of water and to drink until they reach repletion (*Figure 1A*). We developed an open-field assay to assess appetitive behaviors, including thirst (*Figure 1B*). Flies are placed into an open arena, allowed to acclimate, and then presented with a small receptacle containing a source of water or food. A water source rapidly sets up a stable 3–6% humidity gradient from the edge to the source at the center of the chamber. Flies may perform a complete series of steps of thirst-driven behavior with an accessible water source, from seeking to achieving repletion, or they can be limited to water seeking by placing a mesh grid over the source. The longer flies are water deprived, the more quickly they find and occupy the water source (*Figure 1C,D*). Water deprivation also increases locomotor speed (*Figure 1E*). Given a choice, thirsty flies seek water over food, and hungry flies seek food over water, indicating that deprivation state-specific seeking can be readily measured in the open field assay (*Figure 1F*). A combination of hygrosensory and taste cues guide thirsty flies to water. Blocking access to the water source with the mesh grid had no effect on seeking, whereas removal of the third antennal segment that harbors hygrosensory neurons in the sacculus (antennectomy) reduced seeking (*Figure 1G*). Combining

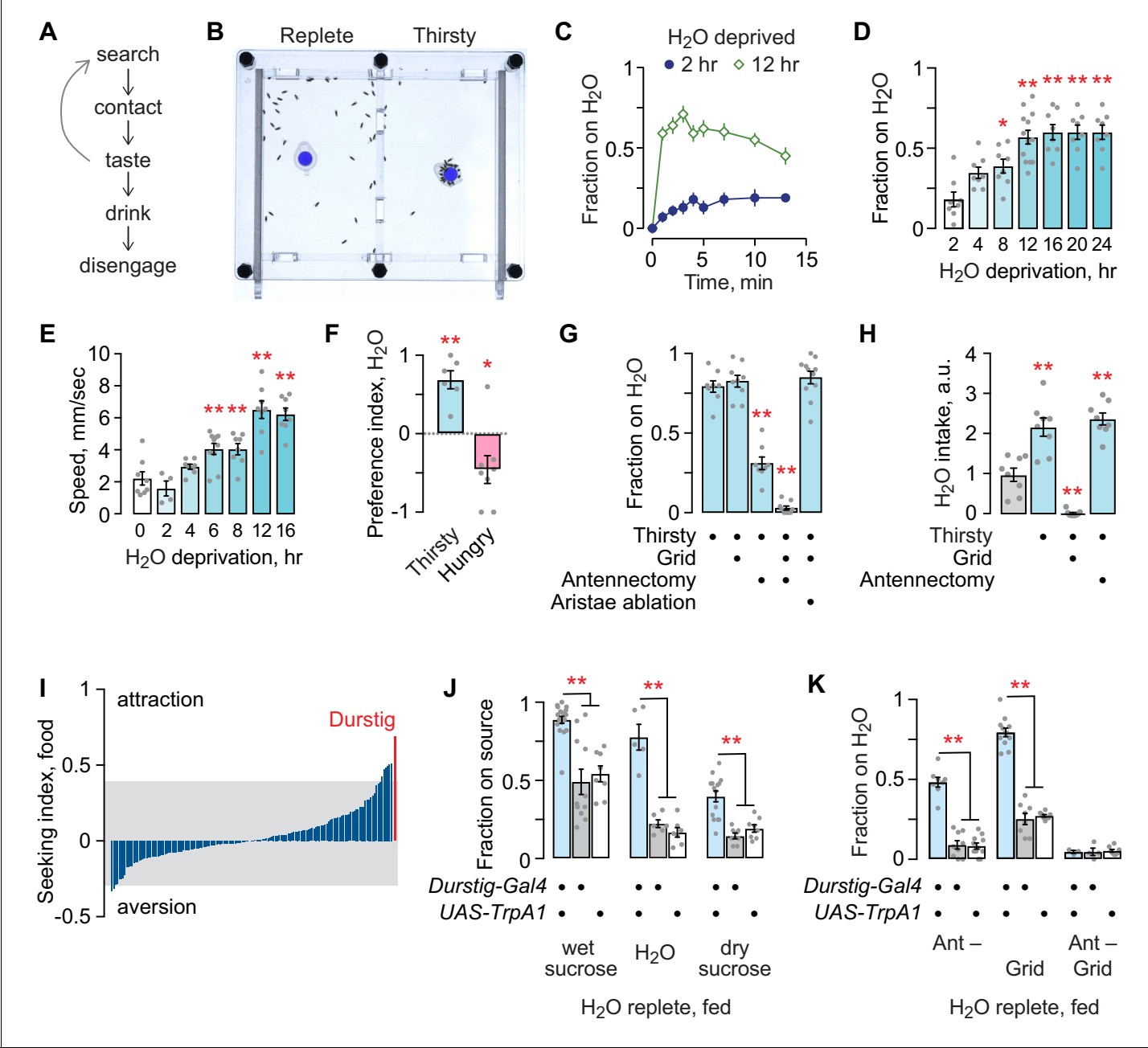

**Figure 1.** Thirst behavior in an open field; a forward screen uncovers Durstig thirst neurons. (**A**) Thirst-induced sequence of behavior. (**B**) Two-chambered open field assay with a vertical divider at the center. Thirsty flies avidly seek and occupy a discrete water source (dyed blue for clarity). (**C**) Occupancy of an open water source over time by a group of 20 flies, n = 8 groups. (**D**) Occupancy increases with increasing water deprivation. One-way ANOVA/Dunnett's compared to 2 hr water deprivation. Each dot represents one group of 20 flies. (**E**) Water deprivation increases locomotor activity. One-way ANOVA/Dunnett's compared to 0 hr. (**F**) Two choice preference for water (1) and dry sucrose (−1) depends on deprivation state. One-sample t-test, compared to 0 (no preference). (**G**) Sensory systems in thirsty water seeking. A mesh grid atop the water source preserves humidity sensing and blocks water contact, taste, and ingestion. Antennectomy, removal of the third antennal segment and aristae, removes humidity and temperature sensors. One-way ANOVA/Dunnett's compared to open/unoperated. (**H**) Sensory input in water intake. Kruskal-Wallis/Dunn's compared to water replete (gray bar). (**I**) Screen for neurons that affect the occupancy of food, using a library of InSITE *enhancer-Gal4* strains driving expression of the TrpA1 heat activated cation channel. Seeking index is *enhancer-Gal4>TrpA1* minus *enhancer-Gal4>+*. Positive index indicates greater occupancy. Gray extends to two standard deviations from the mean. (**J**) Durstig neuron activation drives occupancy of food and water. (**K**) Durstig activation promotes water occupancy through hygrotaxis and contact-dependent mechanisms. Statistics are one-way ANOVA/Tukey's or Kruskal-Wallis/Dunn's, unless indicated otherwise.

The online version of this article includes the following figure supplement(s) for figure 1:

*Figure 1 continued on next page*

*Figure 1 continued*

**Figure supplement 1.** Effects of activating neurons implicated in feeding, reward, and water learning on water seeking.

antennectomy with the mesh grid completely blocked seeking. Antennectomy also removes the aristae that sense temperature; ablation of the aristae alone had no effect on seeking to the inaccessible water source (*Budelli et al., 2019*). Water deprivation increased water intake: removing the third antennal segment had no effect, whereas, as expected, blocking access with the mesh grid fully blocked intake (*Figure 1H*). Thirsty flies were not attracted to an empty receptacle (not shown). Thus, thirsty flies use hygrosensory and taste environmental cues to locate and drink water, and we can isolate the hygrotactic water seeking step from other thirst-related behaviors.

To identify neurons that promote seeking behavior, we acutely activated neurons in 154 different InSITE *enhancer-Gal4* patterns in water replete, well-fed (satiated) flies (*Gohl et al., 2011*). We initially screened for feeding behavior by presenting flies with a discrete source of standard fly food. The screen revealed patterns of neurons that, when activated by the heat-sensitive TrpA1 cation channel, promote or inhibit foraging (*Figure 1I*). One activation pattern, *Durstig-Gal4* (the German for 'thirst'), resulted in markedly high attraction to food. We asked what component of the food source the *Durstig>TrpA1* flies were seeking. Well fed, water replete *Durstig>TrpA1* flies avidly occupied standard fly food, wet sucrose or yeast alone in 1% agar, dry sucrose, 1% agar alone, or water alone, but they were not attracted to empty receptacles (*Figure 1J*). Hygrotaxis toward inaccessible water was strongly potentiated by Durstig activation (*Figure 1K*). Thus, Durstig contains neurons that promote water seeking. We chose to focus on water seeking behavior because it is a specific step in the thirst behavioral sequence that is strongly motivated.

We asked if neurons previously implicated in intake (food and water), water reward memory, and water sensing increase water seeking when activated. Activation of dopamine neurons for reward and aversion had either no effect or suppressed water seeking (*Figure 1—figure supplement 1A*). Moreover, dopaminergic neurons previously implicated in humidity preference and water reward, in the R48B04 pattern, are different from the Durstig water seeking neurons (*Figure 1—figure supplement 1B*). Of neurons implicated in feeding, activation of only two patterns increased water seeking: the NP883 pattern containing the Fdg neurons that promote proboscis extension and liquid food intake, and the R65D05 pattern that contains AstA neurons that inhibit food intake (*Figure 1—figure supplement 1C*; *Chen et al., 2016*; *Flood et al., 2013*; *Hentze et al., 2015*; *Hergarden et al., 2012*; *Pool et al., 2014*). Activation of internal osmosensory ISNs (R34G02), hygrosensory neurons (Ir40a, Ir68a), and water taste neurons (ppk28) did not increase water seeking (*Figure 1—figure supplement 1D*; *Enjin et al., 2016*; *Jourjine et al., 2016*; *Knecht et al., 2016*). These findings reveal segregation of water seeking from many of the previously identified neurons involved in steps of feeding and intake. *Durstig-Gal4*-driven GFP revealed that a complex pattern of neurons is labeled in the brain (*Figure 1—figure supplement 1E*). Because of this, we sought to isolate the individual neurons responsible for water seeking.

## Pattern refinement reveals a small group of neurons that promote thirsty water seeking

We used genetic intersectional techniques to isolate the water seeking neurons embedded in the Durstig pattern. First, subtracting R65D05 neurons from Durstig reduced activation-dependent water seeking, indicating that the water seeking neurons in Durstig and R65D05 are the same (*Figure 2A*). Activation of central brain R65D05 neurons promoted water seeking, whereas their inactivation decreased seeking, indicating that central brain neurons in the R65D05 pattern (and not ventral nervous system neurons or other non-neuronal cells) are both necessary and sufficient for water seeking (*Figure 2B,C*; *Figure 2—figure supplement 1A–E*). Feeding behavior toward dry sucrose was increased with R65D05 inactivation, as expected from previous reports (*Figure 2—figure supplement 1F*; *Chen et al., 2016*; *Hentze et al., 2015*; *Hergarden et al., 2012*). Second, we expressed the GAL4 inhibitor GAL80 in various neurotransmitter classes of neurons and assessed Durstig activation-driven water seeking. Subtracting out neurons with *VGlut-Gal80* (vesicular glutamate transporter enhancer) specifically suppressed Durstig water seeking (*Figure 2—figure supplement 2A,B*). Next, we activated neurons captured by *enhancer-Gal4* transgenes created with

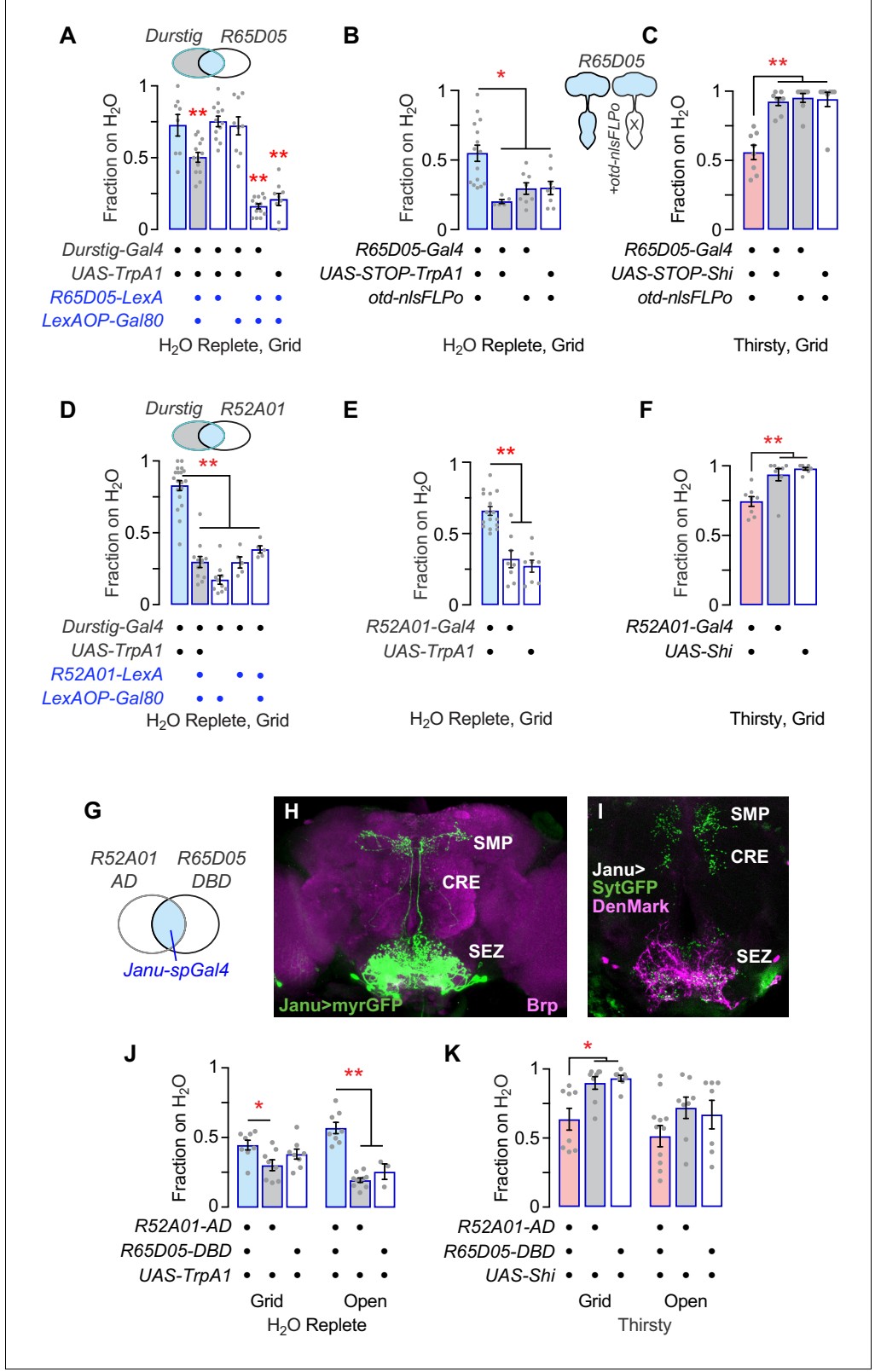

**Figure 2.** Isolation of Janu thirst neurons from the Durstig pattern. (**A**) Subtraction of R65D05 neurons (*R65D05>Gal80*) from the Durstig pattern decreases Durstig activation water seeking. One-way ANOVA/Dunnett's compared to *Durstig>TrpA1*. (**B**) Activation of R65D05 neurons specifically in the central brain promotes water seeking. *otd-nlsFLPo* expresses Flippase in the central brain and not the ventral nervous system, to remove the
*Figure 2 continued on next page*

*Figure 2 continued*

STOP cassette from *UAS-STOP-TrpA1*, thus restricting TrpA1 expression to central brain R65D05 neurons. (**C**) Inactivation of R65D05 neurons specifically in the central brain inhibits thirsty water seeking. (**D**) Subtraction of R52A01 neurons (*R52A01>Gal80*) from the Durstig pattern decreases Durstig activation (*Durstig>TrpA1*) water seeking. One-way ANOVA/Dunnett's compared to *Durstig>TrpA1*. (**E**) R52A01 neuron activation increases water seeking. (**F**) R52A01 inactivation decreases thirsty water seeking. (**G**) A *split-Gal4* (*spGal4*) with *R52A01-AD* and *R65D05-DBD* forms functional GAL4 (*Janu-spGal4*) selectively in the *R52A01/R65D05* overlap. (**H**) Expression pattern of *Janu-spGal4* in the brain, counterstained with the Brp synaptic marker. (**I**) Janu neuron polarity, revealed by presynaptic synaptotagmin-GFP (*UAS-sytGFP*, green) and dendrite localized *UAS-DenMark* (magenta). (**J**) Janu neuron activation increases water seeking. (**K**) Janu neuron inactivation decreases thirsty water seeking. Statistics are one-way ANOVA/Tukey's or Kruskal-Wallis/Dunn's, unless indicated otherwise.

The online version of this article includes the following figure supplement(s) for figure 2:

**Figure supplement 1.** Characterization of *R65D05-Gal4* neurons for thirst and hunger behaviors.
**Figure supplement 2.** Characterization of *R52A01-Gal4* neuron for thirst behaviors.
**Figure supplement 3.** Janu neuron water seeking and feeding behavior characterization.
**Figure supplement 4.** Anatomy of individual Janu neurons.

---

enhancer fragments taken from the VGlut locus, and we identified *R52A01-Gal4* neurons as being able to promote water seeking (*Figure 2—figure supplement 2C,D*). Subtracting R52A01 neurons from the Durstig pattern blocked Durstig activation from increasing water seeking, indicating that Durstig water seeking neurons were also found in R52A01 (*Figure 2D*). Like R65D05, R52A01 activation increased and R52A01 inhibition decreased water seeking (*Figure 2E,F*). Subtracting out neurons with *VGlut-Gal80* decreased thirsty water seeking in both TrpA1-activated R52A01 and R65D05, suggesting that the water seeking neurons were shared between these two Gal4 driver transgenes (*Figure 2—figure supplement 2E,F*). To test if R65D05 and R52A01 labeled the same water-seeking neurons, we assembled a *split-Gal4* (*spGal4*) derived from these two enhancer fragments that we named Janu (the Estonian for thirst) (*Figure 2G*). *Janu-spGal4* is *R52A01-Gal4.AD; R65D05-Gal4.DBD*. *Janu-spGal4* is expressed in four bilaterally symmetric neurons in the central brain (*Figure 2H*). Janu neuron postsynaptic inputs are confined to the subesophageal zone (SEZ), and their presynaptic outputs are distributed between the SEZ, the crepine region (CRE), and the superior medial protocerebrum (SMP) (*Figure 2I*). TrpA1 activation of Janu neurons increased water seeking, albeit to a lesser extent than either parental Gal4 driver alone (*Figure 2J*). Conversely, acute inactivation of Janu neurons decreased thirsty water seeking (*Figure 2K*). The behavioral effects of both Janu activation and inactivation on water seeking mapped to the central brain, similar to the parental *enhancer-Gal4s* for *Janu-spGal4* (*Figure 2—figure supplement 3A–C*). Interestingly, feeding behavior was increased when Janu neurons were acutely inactivated, the opposite effect as compared to water seeking (*Figure 2—figure supplement 3D,E*). Thus, the Janu neurons promote water seeking and inhibit feeding behavior.

## GABAergic local interneurons in the SEZ promote water seeking

To identify the specific neurons that control water seeking and feeding behavior, we characterized the individual Janu neurons. First, we used a stochastic labeling technique to determine the anatomy of each Janu neuron (*Figure 2—figure supplement 4*). Two of the four central brain Janu neurons project locally in the SEZ, and the other two neurons project from the SEZ to the CRE and/or the SMP. Second, we determined Janu neurotransmitter/neuromodulator identity using immunohistochemistry with antibodies for specific neurotransmitter classes of neurons and for neuropeptides. We found that the two local SEZ neurons were GABAergic, expressing the VGAT vesicular transporter for GABA; we named them Janu-GABA1 and Janu-GABA2 (*Figure 3A–H*). VGAT precisely localized with nearly all Janu-GABA axonal varicosities, and did not localize with other Janu neurons. Knockdown of VGAT with either of two distinct RNAi transgenes in the Janu neurons decreased water seeking in thirsty animals (*Figure 3I*; *Figure 3—figure supplement 1A*). Knockdown of *Gad1*, encoding the GABA biosynthetic enzyme Glutamic acid decarboxylase 1, in Janu neurons also decreased thirsty water seeking (*Figure 3J*). To ask if Janu-GABA neurons were responsible for promoting water seeking when Janu neurons were activated in water replete flies, we knocked down Gad1 while simultaneously activating the Janu neurons with TrpA1. This manipulation blocked

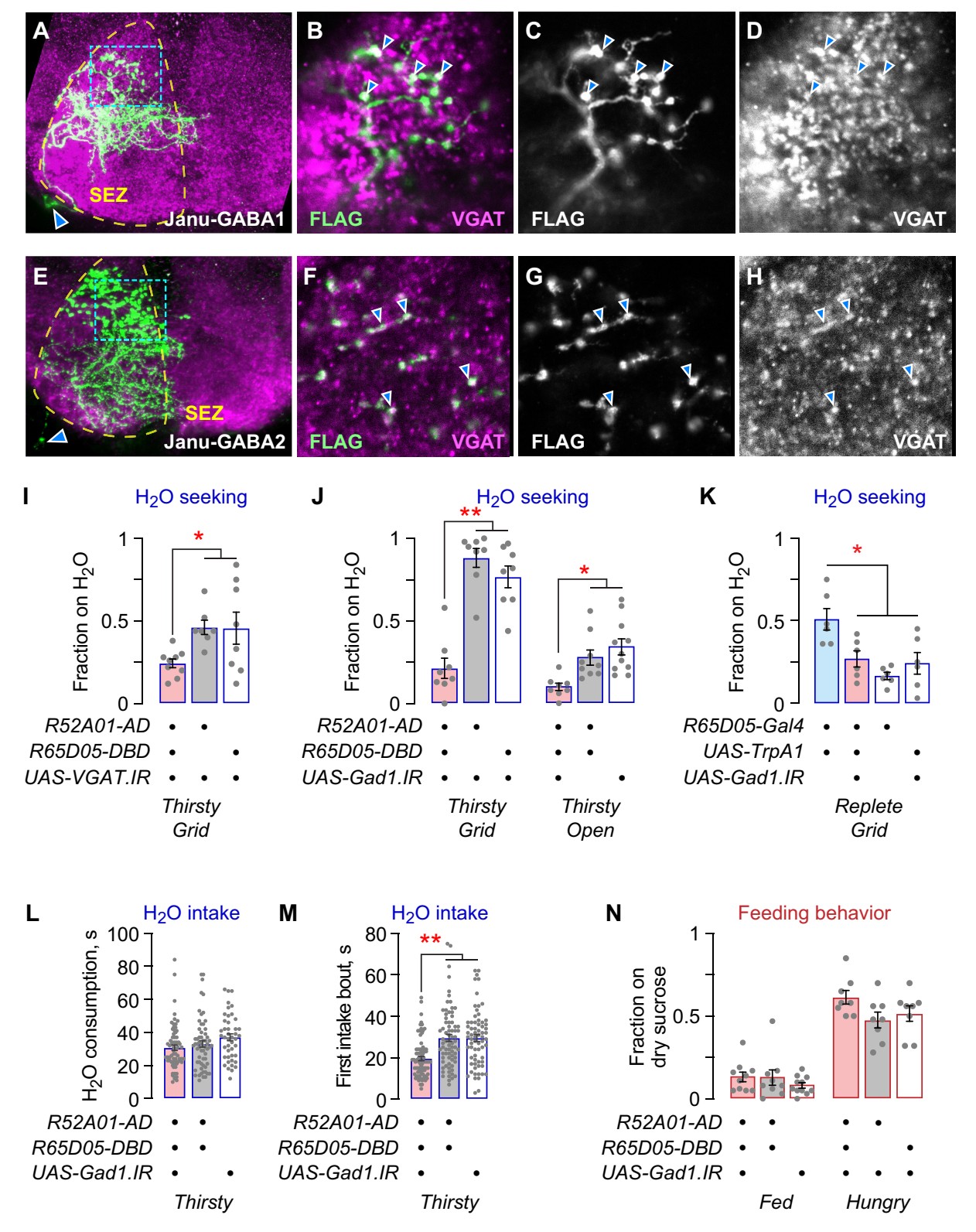

**Figure 3.** Janu-GABA SEZ local interneurons specifically promote water-seeking behaviors. (A–D) Janu-GABA1 interneuron morphology in the SEZ of a Janu>multicolor flip-out brain, compressed z-stack. Arrowhead in (A) points to the cell body. Boxed area is enlarged in (B–D), showing colocalization of the Janu-GABA1 neuron (anti-FLAG) with immunoreactivity for VGAT (arrowheads in B–D). Single 1 μm confocal section. (E–H) Janu-GABA2 interneuron morphology in the SEZ of a Janu>multicolor flip-out brain, compressed z-stack. Arrowhead in (E) points to the cell body. Boxed area is enlarged in (F–

*Figure 3 continued on next page*

Figure 3 continued

H), showing colocalization of the Janu-GABA2 neuron (anti-FLAG) with immunoreactivity for VGAT (arrowheads in **F–H**). Single 1 μm confocal section. (**I**) *VGAT* knockdown in Janu neurons decreases thirsty water seeking. (**J**) *Gad1* knockdown in Janu neurons decreases thirsty water seeking. (**K**) Simultaneous TrpA1 neuronal activation and *Gad1* knockdown in R65D05 neurons blocked *R65D05>TrpA1* increased water seeking in replete flies. (**L**) No effect on water consumption time with *Gad1* knockdown in Janu neurons in thirsty flies. (**M**) Decreased time to termination of first drinking bout with *Gad1* knockdown in Janu neurons in thirsty flies. (**N**) No effect on feeding behavior with *Gad1* knockdown in Janu neurons. Statistics are one-way ANOVA/Tukey's or Kruskal-Wallis/Dunn's.

The online version of this article includes the following figure supplement(s) for figure 3:

**Figure supplement 1.** Additional characterization of Janu-GABA neuron function in thirst behaviors.

activation-dependent water seeking (*Figure 3K*). Thus, the Janu-GABA neurons promote thirsty water seeking and are capable of driving water seeking in water replete animals. To ask if Janu-GABA neurons are specific to the seeking phase of thirst or if they drive a sequence of thirst behaviors, we tested water intake. We presented thirsty flies with a drop of water and measured total water consumption time, the length of the first intake bout, and pharyngeal pumping rate. Gad1 knockdown in Janu neurons had no effect on total water consumption time, but it reduced the length of the first intake bout (*Figure 3L,M*). Pharyngeal pumping rate was unchanged in thirsty Janu neuron *Gad1* knockdown flies, indicating that total intake and motor behaviors were unaffected (*Figure 3—figure supplement 1B,C*). Finally, feeding behavior was unaltered in the Janu Gad1 knockdown flies (*Figure 3N*). Thus, the Janu-GABA neurons are thirst-driven water seeking inhibitory local interneurons in the SEZ.

## AstA neurons reciprocally regulate water seeking and feeding behavior

The Janu-AstA neuron is a bilaterally symmetric interneuron that projects ipsilaterally from the SEZ to the ventral medial SMP, and that expresses the neuropeptide AstA (*Figure 4A–G*; *Figure 4—figure supplement 1A, B*). AstA knockdown in *Janu-spGal4* neurons with either of two independent RNAi transgenes reduced thirsty water seeking (*Figure 4H,I*; *Figure 4—figure supplement 1C–E*). Moreover, loss-of-function AstA mutant flies showed markedly reduced water seeking, both as a homozygote and when in trans to a small chromosomal deletion at the AstA locus (*Figure 4J,K*). To ask if Janu-AstA neurons were responsible for promoting water seeking when Janu neurons were activated in water replete flies, we knocked down AstA while simultaneously activating the Janu neurons with TrpA1. This manipulation blocked activation-dependent water seeking (*Figure 4L*; *Figure 4—figure supplement 1F*). Thus, like Janu-GABA neurons, the Janu-AstA neurons promote thirsty water seeking and are capable of driving seeking in water replete flies. This finding suggests that there may exist a functional relationship between the Janu-GABA and the Janu-AstA neurons. However, there are key functional differences between the Janu-GABA and Janu-AstA neurons. AstA knockdown in Janu neurons did not affect total water consumption time, the length of the first intake bout, pharyngeal pumping rate, or intake measured in the open-field assay (*Figure 4M,N*; *Figure 4—figure supplement 1G–I*). Thus, Janu-AstA neurons are required for thirsty water seeking, and they promote seeking when activated in water replete flies, but they have no role in water intake. Interestingly, AstA in the Janu neurons inhibits feeding behavior in both well-fed and hungry flies (*Figure 4O*; *Figure 4—figure supplement 1J*). Thus, the Janu-AstA neurons oppositely regulate water seeking and feeding behavior. This is in contrast to the Janu-GABA neurons that are specific to water-seeking behavior.

## Neural activity and motivational properties of Janu neurons

To test if thirst and hunger regulate the activity of the Janu neurons, we used the TRIC genetic calcium indicator that accumulates GFP in active neurons (*Gao et al., 2015*). The Janu-GABA1 neurons were selectively activated in thirsty flies; we detected no thirst-dependent activity in any other Janu neurons (*Figure 5A,B*). Moreover, hunger did not change the activity of any Janu neuron (*Figure 5C*). These data indicate that the Janu-GABA1 neurons selectively increase activity with increasing water deprivation.

Finally, we asked if Janu neuron activity imparts appetitive or aversive value to the fly. We measured positional preference in an open-field arena where the light-gated ion channel Chrimson,

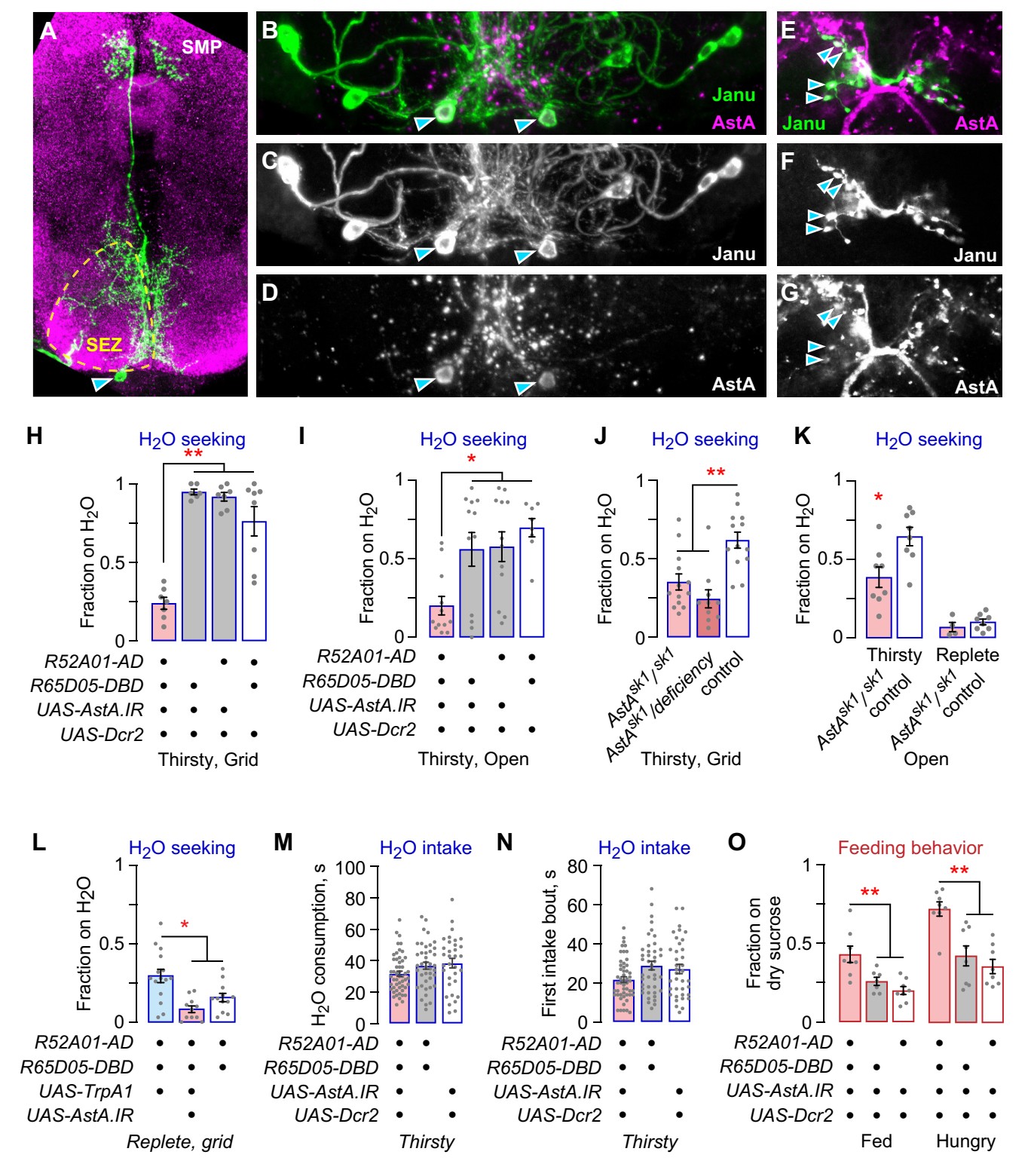

**Figure 4.** The Janu-AstA neuron reciprocally regulates thirst and feeding behaviors. (A–G) Janu-AstA neuron morphology and AstA expression. (**A**) Janu-AstA neuron morphology obtained from a multicolor flip-out brain. Green: anti-FLAG; magenta: anti-VGAT. Arrowhead points to cell body. Process at bottom left is from a co-labeled Janu-GABA neuron. (B–D) The Janu-AstA neuron cell body is immunoreactive for AstA. Arrowheads points to the cell body. (E–G) Colocalization of AstA immunoreactivity with Janu-AstA presynaptic endings in the medial SMP. Arrowheads point to a subset of overlapping expression. Single 1 μm confocal section. (**H, I**) *AstA* knockdown in Janu neurons reduced thirsty water seeking and open water occupancy.

*Figure 4 continued on next page*

*Figure 4 continued*

(J, K) Flies lacking *AstA* showed reduced thirsty water seeking and open water occupancy. (K) t-test. (L) Simultaneous TrpA1 neuronal activation and *AstA* knockdown in Janu neurons blocked increased water seeking in replete flies. (M) No effect on water consumption time with *AstA* knockdown in Janu neurons in thirsty flies. (N) No effect on time to termination of first drinking bout with *AstA* knockdown in Janu neurons in thirsty flies. (O) *AstA* knockdown in Janu neurons increased food occupancy in well-fed and hungry flies. Statistics are one-way ANOVA/Tukey's or Kruskal-Wallis/Dunn's, unless indicated otherwise.

The online version of this article includes the following figure supplement(s) for figure 4:

**Figure supplement 1.** Additional characterization of Janu-AstA neurons.

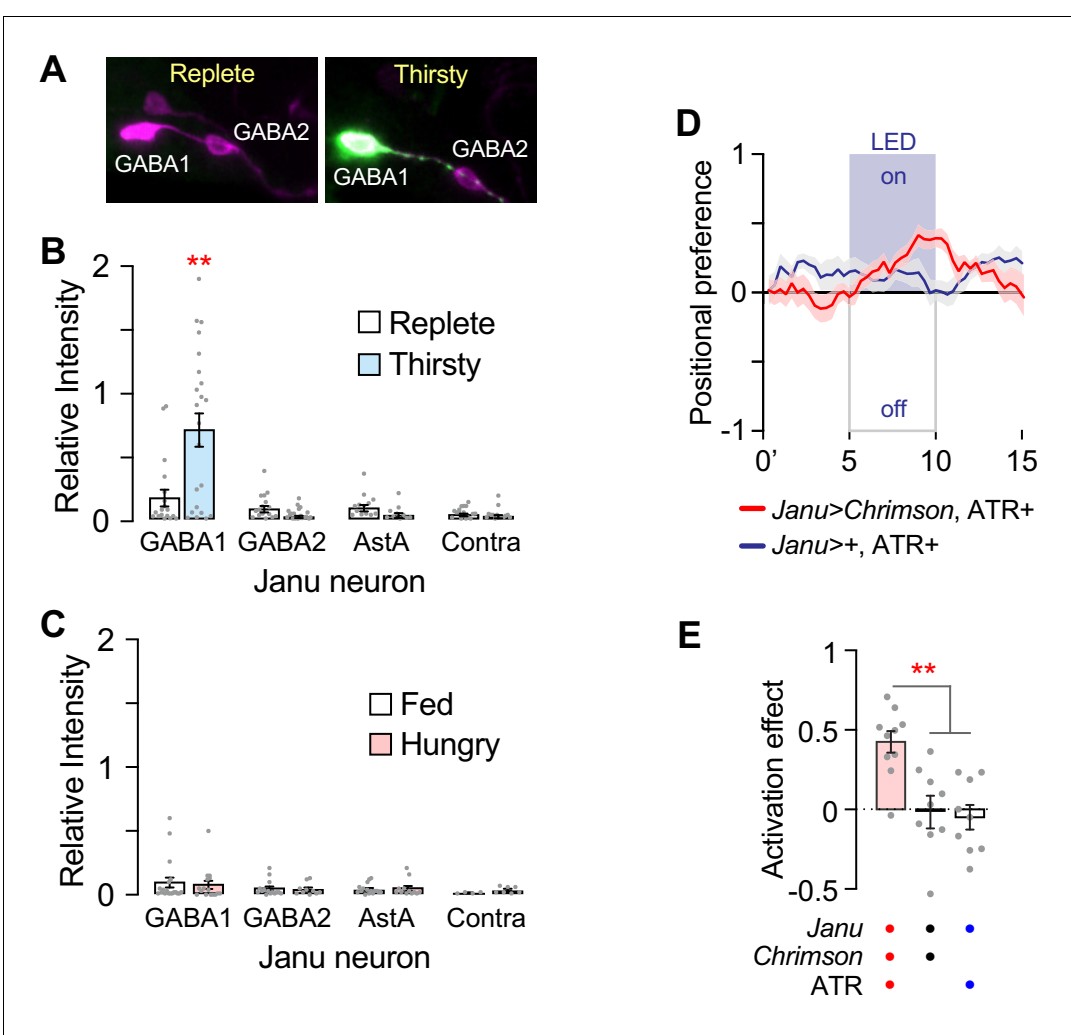

**Figure 5.** Thirst regulation of Janu neuron activity, and positive valuation of optogenetic activation of Janu. (A) Examples of TRIC detection of neuronal activity (green) and the pattern of cells expressing the calcium indicator (magenta) under water replete and thirsty conditions in Janu neurons, detected with anti-GFP (green) and dsRed (magenta) antibodies. (B, C) Relative intensity of TRIC signal (green in A) to pattern (magenta in A) for the individual Janu neurons in replete and thirsty (B), and well-fed and hungry (C) flies. Multiple adjusted t-test, with FDR = 1%. (D) Positional preference of *Janu>Chrimson* (red) and *Janu>+* (blue) water replete and well-fed flies that were fed all-trans retinal (ATR). Positive values indicate attraction to LED-ON half of the rectangular arena, controlled for arena side bias. (E) Quantification of D. Activation effect is the average positional preference for a 1 min bin at 4 min (activation lights off) subtracted from 9 min (activating LEDs on) (see also *Figure 5—figure supplement 1A*). One-way ANOVA/Tukey's.

The online version of this article includes the following figure supplement(s) for figure 5:

**Figure supplement 1.** Flies prefer activation of water seeking neurons.

expressed in Janu neurons, could be selectively activated in either half of an open-field arena. *Janu>Chrimson* flies that were well fed and water replete were attracted to the activation side of the arena (*Figure 5D,E*; *Figure 5—figure supplement 1A*). Flies showed a similar preference for Chrimson activation of the R65D05 pattern that also contains the Janu water seeking neurons (*Figure 5—figure supplement 1B,C*). These findings suggest that water seeking driven by Janu neurons is rewarding. Moving up a humidity gradient when thirsty may imply impending repletion to the fly, and an end to the negative valence state of thirst (*Betley et al., 2015*).

## Janu-AstA neurons are anatomically and functionally upstream of AstA-R2-expressing NPF neurons

NPF promotes feeding behaviors in *Drosophila* (*Shao et al., 2017*; *Shohat-Ophir et al., 2012*; *Wu et al., 2003*). Janu-AstA projects to the SMP, where NPF neurons elaborate synaptic endings. Janu-AstA presynaptic outputs partially overlapped with NPF in the SMP (*Figure 6A,B*). Moreover, synaptic GRASP (GFP Reconstitution Across Synaptic Partners) revealed that Janu-AstA neurons are presynaptic to NPF neurons (*Figure 6C*; *Macpherson et al., 2015*). To test for signaling to NPF neurons by AstA, we reduced expression of the two AstA receptors AstA-R1 and AstA-R2 in NPF neurons using previously characterized RNAi transgenes (*Yamagata et al., 2016*). AstA-R2, but not AstA-R1, knockdown decreased thirsty water seeking, and also increased feeding behavior, similar to AstA function in the Janu-AstA neurons (*Figure 6D*; *Figure 6—figure supplement 1A*). AstA-R2 knockdown in NPF neurons, however, reduced both total water consumption time and the duration of the first intake bout, indicating that AstA signaling to NPF neurons affects both water seeking and water ingestion (*Figure 6E,F*; *Figure 6—figure supplement 1B*). Consistent with AstA-R2 functioning as an inhibitory receptor in NPF neurons, we observed increased water seeking in flies completely lacking NPF and in flies with NPF knocked down in NPF neurons (*Figure 6G,H*). Measurement of neuronal activity in NPF neurons indicated that only two NPF neurons, the P1 neurons and the L1-L neurons, showed any activity, and they both increased activity in thirsty flies (*Figure 6I*; *Figure 6—figure supplement 1C*). These findings indicate that NPF neurons respond to thirst and that NPF neurons receive AstA signals via AstA-R2 to reciprocally regulate thirst and hunger behaviors. *Figure 7* summarizes our current understanding of the thirst circuitry and its relation to feeding behavior.

## Discussion

Thirst is an internal state encoded by discrete neural circuits that generate innate goal-directed behavioral actions. We used an unbiased screen followed by genetic techniques to discover central brain interneurons that regulate specific aspects of thirsty water seeking, and that contribute to a specific motivational state. While sensory and interoceptive inputs are known, there is little understanding of how the brain forms the representation of thirst and coordinates sequences of behaviors to achieve water repletion. The Janu-GABA water seeking and Janu-AstA to NPF water seeking and feeding behavior circuit elements define specific functional steps of thirst representation, and they provide a means to help understand the interactions of different need states.

The Janu-GABA neurons are GABAergic local interneurons whose activity is increased by thirst and that promote thirsty water seeking. GABA in Janu-GABA is sufficient for driving water seeking in the water replete state, and it is necessary for thirsty seeking. The role of the Janu-GABA neurons appears to be limited to the seeking step of thirst behavior, since thirsty water consumption is largely unaffected by their inhibition. The specific role for Janu-GABA in promoting persistence during the first thirsty intake bout may reflect the end of the seeking component of the sequence of thirst behaviors that may overlap with the beginning of the intake phase. Moreover, the Janu-GABA neurons did not affect feeding behavior and are not activated by hunger. Thus, the Janu-GABA neurons increase thirsty seeking to sources of water. The Janu-GABA neurons are functionally distinct from the Fdg local interneurons that are also located in the SEZ (*Flood et al., 2013*; *Pool et al., 2014*). The Fdg neurons increase intake of liquid food and promote proboscis extension, behavioral functions we did not observe for the Janu-GABA neurons. We showed that the Fdg neurons may also promote water seeking. However, we did not identify the specific neurons in the *NP883-Gal4* pattern that are responsible for water seeking. Defining the source of the signal to drive Janu-GABA dependent thirsty seeking and the mechanism of thirsty seeking require further elaboration of the

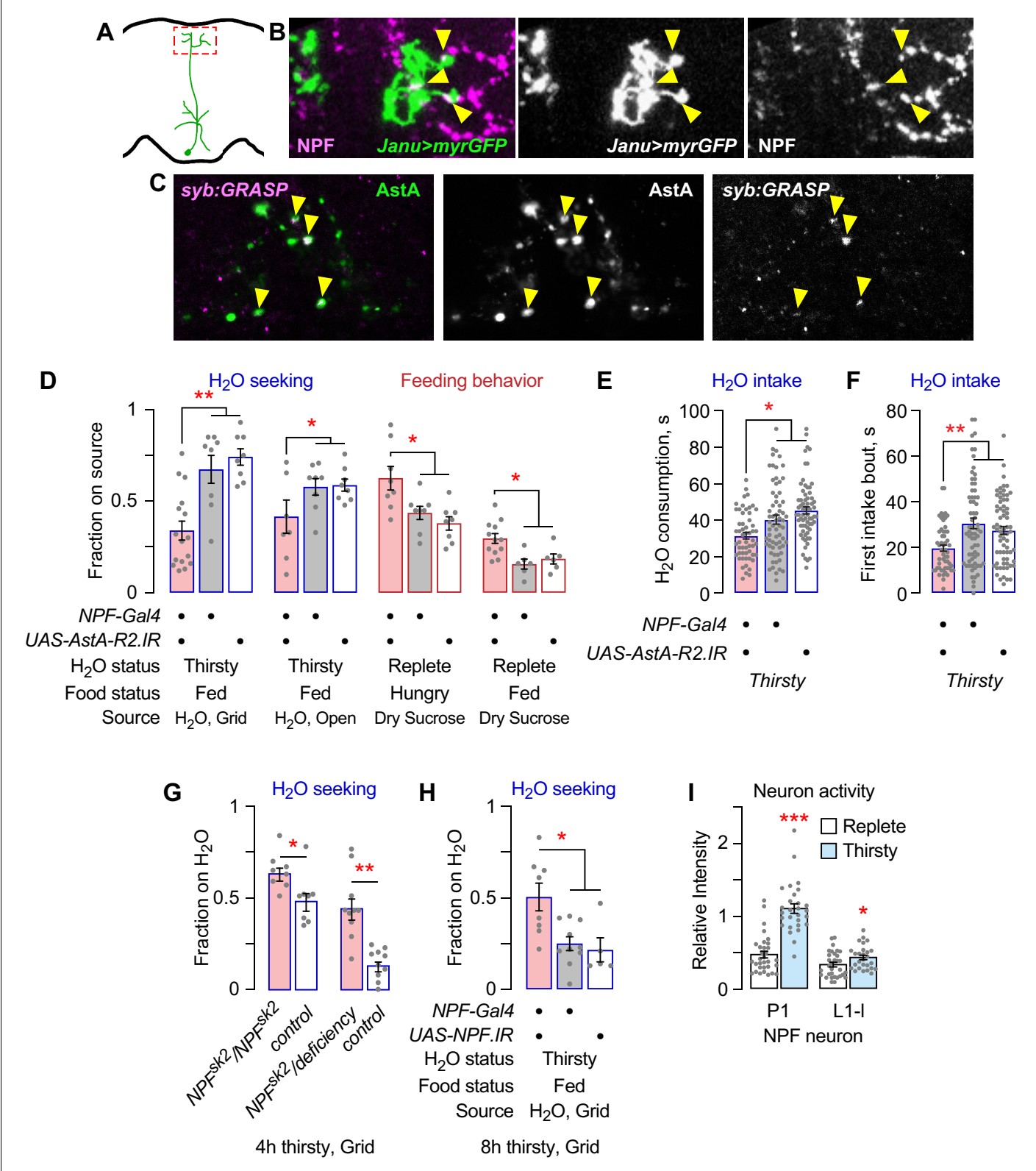

**Figure 6.** NPF neurons are postsynaptic to Janu-AstA neurons via the AstA-R2 receptor, and NPF inhibits water seeking. (**A**) Janu-AstA presynaptic endings in boxed area imaged for immunohistochemistry. (**B**) Janu-AstA presynaptic endings partially overlap with NPF immunolabeling. Single 1 μm confocal section. Yellow arrowheads point to overlap. (**C**) Synaptic GRASP (syb:GRASP) between *R65D05-Gal4>UAS-syb:spGFP[1–10]* and *NPF-LexA>LexAOP-spGFP[11]* in thirsty flies. Single 1 μm confocal section. Yellow arrowheads point to GRASP GFP signal that overlaps with AstA

*Figure 6 continued on next page*

*Figure 6 continued*

immunoreactivity in the superior medial protocerebrum. (**D**) Behavioral effects of *AstA-R2* RNAi in NPF neurons. (**E**) *AstA-R2* knockdown in NPF neurons decreased total water consumption time. (**F**) *AstA-R2* knockdown in NPF neurons decreased the length of the first water intake bout. (**G**) Flies mutant for NPF show increased water seeking when mildly water deprived (4 hr). Mann-Whitney t-test. (**H**) *NPF* knockdown in NPF neurons increased water seeking in mildly water deprived flies (8 hr). (**I**) NPF neuron activity in thirsty flies with TRIC. Activity was detected only in P1 and L1-l NPF neurons. t-test. Statistics are one-way ANOVA/Tukey's or Kruskal/Wallis/Dunn's, unless indicated otherwise.

The online version of this article includes the following figure supplement(s) for figure 6:

**Figure supplement 1.** Additional characterization of the role of NPF neurons in thirst behaviors.

circuitry for thirst and other behaviors in the SEZ region of the *Drosophila* brain. We speculate that the Janu-GABA neurons may function upstream or downstream of the osmosensory ISN neurons that, when inhibited, promote water intake (*Jourjine et al., 2016*). Janu-GABA neuron activation in the SEZ is predicted to lead to inhibition of SEZ neurons, perhaps to repress competing behavioral states represented in the SEZ. Alternatively, the Janu-GABA neurons may repress repressor neurons of thirsty seeking, that remain to be identified.

Janu-AstA is a bilaterally symmetric AstAergic projection interneuron with inputs in the SEZ and outputs in the medial SMP. AstA in the Janu-AstA neuron promotes thirsty water seeking, and it inhibits feeding behavior in a hunger-independent manner. For thirst, AstA in the Janu-AstA neuron was specific to thirsty seeking with no impact on water intake, indicating a functional role similar to Janu-GABA for thirst. However, we did not detect changes in Janu-AstA neuronal activity with either thirst or hunger. This lack of response may indicate a tonic role for AstA in feeding and thirst. Tonic release of neuropeptides can tune neural circuit function, including the peptidergic modulation of hunger (*Fu et al., 2004*; *Tu et al., 2005*). Peptidergic signaling can also be tuned by changing the amount of the receptor in the receiving neuron, as was demonstrated for sensory tuning by the short-NPF (sNPF) neuropeptide (*Root et al., 2011*). Interestingly, knockdown of either AstA or GABA in the Janu neurons blocked water seeking evoked by Janu activation. Janu-GABA and Janu-AstA may work in series in the same circuit, or in parallel in separate circuits that converge, to control thirsty-seeking behaviors.

Whereas Janu-GABA and Janu-AstA may work together for thirsty seeking, Janu-AstA also regulates feeding behavior, indicating branched functionality in water seeking circuitry. Janu-AstA transmits thirst and hunger information to NPF neurons through the AstA-R2 receptor on NPF neurons, where NPF coordinately regulates thirsty seeking and feeding behavior. NPF is well studied in *Drosophila* and in vertebrates as a promoter of feeding behavior – our work indicates a reciprocal role

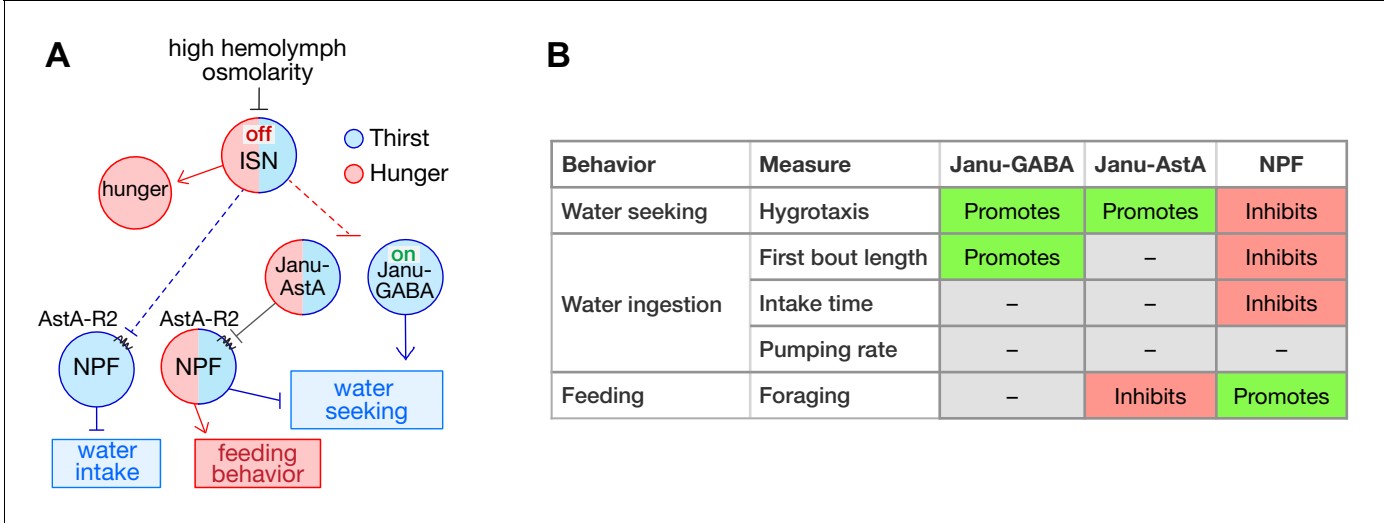

**Figure 7.** Thirst interneuron circuitry in *Drosophila*. (**A**) Functional relationship for thirst and feeding behavior. The dotted lines represent putative connectivity between interoceptive sensory neurons (ISN) and targets neuropeptide F (NPF) and Janu neurons. (**B**) Functions of GABA, AstA, and NPF in their respective thirst interneurons.

for NPF in thirst behavior (*Chen et al., 2019*; *Luquet et al., 2005*; *Pool and Scott, 2014*; *Shen and Cai, 2001*). Coordinated regulation of thirst and hunger is most clearly manifested behaviorally as prandial thirst (thirst induced by feeding) and as dehydration anorexia (thirst suppression of feeding) (*Fitzsimons, 1972*; *Zimmerman et al., 2017*). Janu-AstA and AstA-R2-positive NPF neurons join a small list of neurons that oppositely regulate thirst and hunger behaviors: the osmosensory ISN neurons oppositely regulate food and water intake, and Ion Transport Peptide (ITP) neurons promote thirst and inhibit food intake (*Gáliková et al., 2018*; *Pool et al., 2014*). It remains unclear if all these neurons support water seeking, or if they function specifically during ingestion.

Interestingly, two NPF neurons, the P1 and L1-l neurons, showed increased activity in thirsty flies. Hunger also selectively increases the activity of these two NPF neurons (*Beshel and Zhong, 2013*). This might indicate that the P1 and L1-l represent an internal state imbalance rather than a specific internal state like hunger or thirst. Additionally, the direction of their regulation – increased activity when thirsty – is opposite of what is expected for NPF in these neurons to inhibit thirsty seeking. Thus, we suggest that Janu-AstA may synapse onto different NPF neurons that, like Janu-AstA, do not exhibit state-dependent changes in neuronal activity, but that promote thirst and inhibits feeding behavior. A potential mechanism may be through NPF-NPF neuronal synapses: both the P1 and the L1-l neurons express the NPF receptor NPFR (*Shao et al., 2017*). Moreover, it is likely that one or more of the water-seeking circuit elements we discovered feeds into navigational decision making circuitry to direct hygrotaxis toward a water source and to suppress other goal directed behavior in a dynamic fashion (*Hulse et al., 2020*; *Lyu et al., 2020*).

Optogenetic activation of the Janu or the R65D05 neurons induces place preference, indicating that animals 'like' activation of these neurons. Thirst and hunger are deprivation states that have a negative valence, therefore the Janu neurons likely do not encode the state of deprivation (*Burnett et al., 2016*; *Leib et al., 2017*; *Sutton and Krashes, 2020*). We propose that, instead, they encode the anticipation of terminating a thirsty state. To a thirsty fly, the presence of a humidity gradient likely predicts impending water intake and repletion, which may have rewarding value.

## Materials and methods

### Strains and culturing

All strains were outcrossed for five generations to the Berlin genetic background. Flies were raised on standard food containing agar (1.2% w/v), cornmeal (6.75% w/v), molasses (9% v/v), and yeast (1.7% w/v) at 25°C and 60% humidity in a 16:8 light:dark cycle. For experiments with *UAS-Shibire^ts* and *UAS-TrpA1*, flies were reared and held at 18–23°C prior to testing. For *split-Gal4*, RNAi, and *UAS-STOP-effector*/FLP experiments, adult F1 progeny were held at 29°C for 2–3 days prior to testing to enhance transgene activity. *Supplementary file 1* provides a list of the *InSITE-Gal4* strains screened by neuronal activation for feeding behavior, and their seeking index (fraction on food, *InSITE-Gal4>TrpA1* – *InSITE-Gal4>+*) (*Gohl et al., 2011*). *Durstig-Gal4* is an *InSITE-Gal4* enhancer trap inserted into the first intron of *CG4502* on chromosome 2. All strains used in this study are described in *Supplementary file 2*.

### Behavioral measurements

Genotype and treatment were blinded for all behavioral counts. Groups of approximately 20 adult males were collected 1–2 days prior to the experiment. One group is an n = 1. Matched experimental and control groups were tested within day, and experiments were performed across days, to account for between-day variability. For water deprivation, flies were placed in empty culture vials containing 1 mL 5% sucrose that was dried onto a 2.5 × 15 cm strip of Whatman 3 MM filter paper. For food deprivation, flies were placed into empty culture vials containing water saturated 3 MM filter paper. Dry, rough cut cane sugar (La Perruche Pure Cane Rough Cut Cubes) was used in all food seeking experiments, except for the original screen that used standard fly food. Sources of food or water were presented to flies either on a small square of Parafilm or in the cap of a 0.65 mL microcentrifuge tube. Water was made inaccessible by gluing a 210–300 μm mesh grid onto a cap. Flies were deprived of food or water for the times indicated in *Supplementary file 3*. Thin-walled Plexiglas behavioral chambers were designed with two side-by-side arenas, each arena measuring 45 × 75×10 mm. Chambers were designed and built by IO Rodeo (Pasadena, CA); design files are

available from the company. The chambers and sources were acclimated to the testing temperature in a Peltier incubator, and flies were acclimated into the chambers for 10 min prior to introducing the source (IN45, Torrey Pines Scientific). For TrpA1 and Shibire thermogenetics, flies were temperature acclimated for 20 min before being introduced into the chamber. Humidity was measured using an EK-H4 multiplexer with SHT71 sensors (Sensirion). The incubator maintained 30–40% relative humidity. For thermogenetic activation experiments with UAS-STOP-TrpA1-mCherry, flies were pre-acclimated for 20 min in food or deprivation vials, then 10 min in the behavioral arena, as previously reported (*Asahina et al., 2014*). Flies were filmed from above with Logitech C510 web cameras at 10 fps with the arena placed on white light LED panel (Edmund Optics). The number of flies on the source was divided by total number of flies to determine the fraction of flies on the source. In binary choice experiments, dry sucrose and water were placed in closely apposed receptacles just prior to performing the experiment.

Locomotor speed was calculated per group as the total distance traveled divided by the total time each individual was tracked for 10 s intervals every 5 min, and then averaged across the 10 min and 15 min time points, allowing for acclimation to the arena (*Wolf et al., 2002*).

To measure intake in the open-field arena, water was made to 0.2% w/v erioglaucine and presented to flies for 20 min in the behavioral chamber. Consumption was determined spectrophotometrically in cleared fly extracts at 628 nm by a BioTek Epoch two microplate reader, background subtracted with flies fed undyed water, and was normalized to the *enhancer-Gal4>+* control strain.

Intake time and pumping rate were measured essentially as previously described (*Manzo et al., 2012*). Flies were glued to slides with clear nail polish while cold anesthetized and allowed to recover at 60% humidity and 25℃ for 1–2 hr. Consumption time was measured by repeatedly presenting the proboscis with a water drop extruded from the end of a 5 μL calibrated micropipet (Avantor), until flies failed to extend the proboscis to four sequential presentations. The first bout length was the time from first engagement to first complete disengagement with the water drop. Pumping rate was determined with manual counts of video recordings of pharyngeal contractions during the initial 10 s of engagement, or shorter if the flies disengaged earlier.

We used a custom-built device for determining the positional preference of optogenetically stimulated flies (Flidea). Groups of about 10 flies were kept in the dark on standard food containing 35 mM all-trans retinal for 2–4 days (Sigma). They were placed in an 80 × 40 × 3 mm arena with a clear roof and white acrylic floor. Optogenetic LEDs (617 nm) were arrayed 1 cm below the arena and were switched to allow illumination of either 40 × 40 mm side of the arena. Optogenetic stimulation was at 40 Hz for 200 ms every second. Flies were allowed to acclimate to the arena, and then were given a 5 min off, 5 min on, 5 min off pattern of stimulation, with the ON side randomly assigned. Filming was done at 2 fps in dim ambient light. Object recognition and position in the arena was determined using custom developed DTrack software (*Ro et al., 2014*). Fly positional preference was calculated as the number of flies on a side divided by the total number of flies. Experiments where flies showed a strong side bias prior to optogenetic stimulation were excluded from further analysis. The resulting positional information was averaged into 20 s bins. The activation effect is the positional preference averaged across minute nine minus across minute 4, such that a positive value indicates increased occupancy of the LED-on side when the LEDs are on. The experiment was repeated with flies from 5 to 8 separate crosses across different days.

Statistical measurements were made with Prism 8.0 (GraphPad). One-way ANOVA followed by Tukey's post-hoc comparisons (or Bonferroni post-hoc planned comparison) were used when data did not show unequal variance by the Brown-Forsythe test, otherwise the Kruskal-Wallis test followed with Dunn's post-hoc was used. t-tests were two-tailed (*Supplementary file 3*). Error bars are the SEM. Dots overlaid on the bar graphs are the value measured for individual groups. No statistical methods were used to predetermine sample sizes. Sample sizes are similar to those used in other *Drosophila* behavioral experimental paradigms, and they were held constant with rare exceptions. No data was excluded using statistical methods.

## Immunohistochemistry

Adult fly brains were dissected in PBS with 0.1% Triton X-100 (PBT), fixed overnight at 4℃ in PBT with 2% paraformaldehyde, blocked in PBT with 5% normal goat serum and 0.5% bovine serum albumin, and immunostained as described previously (*Kong et al., 2010*). Antibodies were rabbit anti-GFP (1:1000, Life Technologies), chicken anti-GFP (1:1000, AbCam), mouse anti-GFP for GRASP

(1:100, Sigma), rabbit anti-dsRed (1:500, Takara Bio), mouse anti-Bruchpilot (1:25, Developmental Studies Hybridoma Bank, Iowa), rabbit anti-VGAT (1:200, gift from David Krantz, UCLA), rabbit anti-AstA (1:2000, Jena Bioscience), rabbit anti-NPF (1:100, RayBiotech), and rat anti-FLAG (1:200, Novus Biologicals). Samples were mounted in Vectashield (Vector Laboratories) and imaged on a Zeiss LSM-880 confocal microscope. Image stacks were processed in Fiji, and brightness and contrast were adjusted in Photoshop CC 2021 (Adobe).

# Acknowledgements

We thank Thomas Clandinin, Daryl Gohl, and Marion Silies for InSITE strains, Flidea for building the custom optogenetic device, Ashley Smart and Thomas Clandinin for imaging help, Anand Bala Subramaniam for access to the LSM-880 confocal microscope, and Donnoban Orozco, Nailah Fatima, Natalie Banh, Winnie Wu, Lawrence Fung, Kenneth Mata, Maurice Jackson, Van Thai, and Layne Hazam for help with the experiments. This work was supported by UC Merced startup funds to FWW.

# Additional information

## Funding

| Funder | Grant reference number | Author |
| --- | --- | --- |
| UC Merced | Startup funds | Fred W Wolf |

The funders had no role in study design, data collection and interpretation, or the decision to submit the work for publication.

## Author contributions

Dan Landayan, Conceptualization, Formal analysis, Investigation, Methodology, Writing - original draft, Writing - review and editing; Brian P Wang, Software, Formal analysis, Investigation, Methodology, Writing - review and editing; Jennifer Zhou, Investigation; Fred W Wolf, Conceptualization, Data curation, Software, Formal analysis, Supervision, Funding acquisition, Investigation, Visualization, Methodology, Writing - original draft, Writing - review and editing

## Author ORCIDs

Fred W Wolf  https://orcid.org/0000-0002-4230-194X

## Decision letter and Author response

Decision letter https://doi.org/10.7554/eLife.66286.sa1
Author response https://doi.org/10.7554/eLife.66286.sa2

# Additional files

## Supplementary files
• Supplementary file 1. InSITE strains and seeking index.
• Supplementary file 2. Strains and resources.
• Supplementary file 3. Experimental conditions and statistical analyses.
• Transparent reporting form

## Data availability
All data generated or analysed during this study are included in the manuscript and supporting files.

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
