## [Decision Letter]

**Acceptance summary:**

The work identifies multiple types of "Janu" neurons in *Drosophila* with specific anatomical and neurochemical properties that control the thirst reponse as well as other unique (but related) behaviors. The Janu-AstA neuron is argued to synapse with and signal to NPF neurons, and evidence is provided supporting the role of this neuronal pathway in water-seeking. These interesting set of neurons constitute a valuable resource for further understanding of thirst-induced behaviours.

**Decision letter after peer review:**

[Editors’ note: the authors submitted for reconsideration following the decision after peer review. What follows is the decision letter after the first round of review.]

Thank you for submitting your work entitled "Allatostatin A thirst interneurons promote water seeking and limit feeding behavior in *Drosophila*" for consideration by *eLife*. Your article has been reviewed by 3 peer reviewers, and the evaluation has been overseen by Mani Ramaswami as Reviewing Editor and K. VIjayRaghavan as Senior Editor.

All three reviewers expressed strong interest and enthusiasm for the work. However, as you will read, there was also a consensus that the considerable amount of extra work on presentation and content necessary before potential acceptance would require more than the two months allowed for revisions to a paper in *eLife*. For this and other reasons listed below, after extensive discussion and consultation among the reviewers, we regret to inform you that this paper will not be considered further for publication in *eLife*. The reviewers however, are supportive enough to suggest that, should you choose to do so, then *eLife* will remain willing to examine a carefully revised and resubmitted manuscript as a new submission to the journal.

A single consensus review is provide below, in addition to the individual reviews, and we hope you will find these to be useful.

Summary:

Landayan, Zhou, and Wolf present a new behavioral assay to test different aspects of water/food seeking behaviors in *Drosophila*. From an initial activation screen of 154 Gal4 enhancer trap flies, the work identifies several that mediate food seeking, in particular Durstig-Gal4, where activation of marked neurons causes shows the highest attraction and occupancy level toward food. Significantly, Durstig-Gal4 marked neurons are involved in modulating both food and humidity seeking behaviors. From this starting point, using a vast array of neural activation/inhibition experiments, neuromodulator RNAi, and some neuroanatomy and Split Gal4s, they identify a line called Janu, which labels 4 neurons in the central brain, 3 interneurons in the ventral nervous system and 2 ascending neurons projecting from the ventral nervous system to the central brain. These Janu neurons are either GABAergic or produce allatostatin and may function as a multi-modal integrator required for water seeking (but not water intake). Thus, this subset of inhibitory and Allatostatin A-positive neurons promote thirst-driven behavior. Using synaptic GRASP, they show that Janu-Ast neurons connect to NPF expressing neurons.

Previous work on the interplay between hunger and thirst has focused on sensory circuits, so identifying central neurons mediating thirst would represent a significant contribution to understanding this process. Going from an unbiased behavior screen to a subset of neurons docked into a circuit is admirable. The work is careful, thorough, and on an important topic.

Essential revisions:

1. A major rewrite is required. While the authors may be commended for wrestling with a complex story and for being open about all of the data even when it is hard to reconcile with a simple model, in the end the insights need to be distilled more for public consumption. In current form, the paper is extraordinarily difficult to read and difficult to follow, in part due to the complexity of the methods. There is a large number of different types of behavioral assays (with flies in different internal states), and many different driver lines are presented as the authors home in on those that form the central novel findings (i.e., Janu neurons). The huge amount of data masks the main messages of the paper. There is a lot of chronological and extraneous data included that could be moved to supplementary figures to streamline and we'd strongly suggest the authors prune irrelevant experiments. For example, it jumps from characterization of the new hygrotactic assay (though not the first of this type of assay – – see minor comment below) to a screen using a completely different assay, which gives the impression the authors wanted to show the results of the screen rather than present a logical flow of experiments for the reader. In addition, inconsistent use of nomenclature in text/figures/legends renders it difficult to follow, e.g., Janu neurons are sometimes to referred by name, but often in the behavior data panels only the codes for the split-Gal4s are shown. In general, the data on thirst is stronger than that on NPF and feeding, so the thirst conclusions should be emphasized. One suggestion is to describe the original identification of Durstig neurons as thirst-promoting, then how a subset within that was identified as critical and Allatostatin-A-positive, and then demonstrate that this subset promotes thirst as a drive, suppressing the hunger drive. With a serious organizational revision and streamlining of extra data into supplementary figures, the findings should be clearer and much easier to convey. Some summary schematics could help significantly, particular accompanying the Discussion sectionDiscussion section. For example: one showing the proposed circuit, and a another showing the nested/Venn diagram of Durstig, Janu, AllatostatinA+, and GABA+ neurons, (maybe including their respective GAL4 targeting reagents).

2. One problem than runs through the manuscript is that it remains formally unclear whether the phenotypic differences observed upon activation/inhibition of populations of neurons is due to the difference in their individual roles or to the difference in the magnitude of activation/inhibition. (One might be activating a lot of neurons that have the same role in water seeking and thus the phenotype is stronger than activating only a subpopulation of those neurons). Similarly, many RNAi experiments are presented but, while several do point towards a consistent conclusion, there is no evidence that particular knock-downs work or do not work, and single mutant alleles are shown without rescue or obvious control of genetic background (for both Npf and AstA mutants, a w strain is used as the control). Some of these issues need to be addressed experimentally and certainly in the text.

3. Many of the issues above could be addressed by physiological experiments (calcium imaging or electrophysiology), which would potentially make it easier to appreciate the function of individual neurons. For example, a simple question is whether Janu-AstA and Janu-GABA neurons respond to water vapor and/or water contact? We appreciate such experiments are probably beyond the scope of a reasonable revision. But in the absence of this, extra care should be paid to both establishing the specificity of the genetic perturbations, the resulting phenotypes and in interpretation of the results. As the quantity and complexity of the work makes extraction of a compelling central discovery of broad general interest difficult to appreciate, major revisions should concentrate on marshalling and presenting all the best evidence to arguing for and documenting the identification of thirst-promoting central neurons.

4. The title and the summary are misleading, as the Janu neurons described here are not only allatostatin producing but also GABAergic. Perturbations specific to Allostatin/Janu neurons form only a relatively small part of the entire analysis, which ignores that other Janu neurons are GABAergic. This should be explicitly clarified. Similarly, in the abstract the phrase: "allowed us to uncover the neural basis.……" is a bit of an over-statement (perhaps "a neural basis").

*Reviewer #1:*

Landayan et al. et al. combine a simple behavioral test with an extensive use of genetic tools to identify neurons which promote water seeking and limit feeding behavior in *Drosophila*. Starting from an initial screen of "a" library of 154 Gal4 enhancer trap flies, they select one strain, Durstig, which shows the highest attraction and occupancy level toward food. Starting with this strain, they show that Durstig is involved in modulating both food and humidity seeking behaviors. From this starting point, they analyze which population of neurons are involved in activating Durstig neurons and which neurons are downstream. This demonstration makes an extensive use of various genetic constructions, following a logic that looks solid and supported by behavioral observations, however lacking functional or pharmacological observations. The reasoning is explicitly outlined in the manuscript but the number of constructions used make it difficult to follow to someone which is exterior to this research field.

The authors first verify that the activation of Durstig neurons involves humidity and antennae. Other stimuli related to food seeking or from other receptors are not checked. Then, they proceed to find neurons upstream and downstream of Durstig neurons. They first look at different populations of neurons known to be involved in humidity preference or in the control of liquid or food intake, to see if activating them with TrpA1 (and heat) would induce similar water seeking behaviors.

Since Durstig-gal4 neurons are numerous, the authors tried then to narrow down their analysis using intersectional approaches to find which of these neurons are involved in water-seeking, looking at cholinergic and glutaminergic neurotransmission. This analysis led them to find an enhancer, R52A01, which marks neurons using glutaminergic transmission within the Durstig population of neurons, as well as R65D05 neurons which are involved in inhibiting food intake.

Using a split-Gal4 approach, they could tag neurons (called Janu), namely 4 neurons in the central brain, 3 interneurons in the ventral nervous system and 2 ascending neurons projecting from the ventral nervous system to the central brain.

Using antibodies directed against neurotransmitters and neuropeptides, they show that Janu neurons are either GABAergic or produce allatostatin and that these neurons promote water seeking (but does not change water consumption).

Lastly, using synaptic GRASP, they shown that Janu-Ast neurons connect to NPF neurons.

All in all, this paper sums out an impressive amount of work which leads to the identification and description of neurons involved in water-seeking behavior. To this respect, it constitutes a beautiful analysis. It is a paper difficult to read for me because of the complexity of the methods. I find the title and the summary a bit misleading, as the Janu neurons described here (unless I am mistaken) are not only allatostatin producing but also GABAergic. From the summary and the title, I was also expecting functional studies related to allatostatin.

*Reviewer #2:*

This work presents a new behavioral assay to test different aspects of water/food seeking behaviors in *Drosophila*, and presents a vast array of neural activation/inhibition experiments, neuromodulator RNAi, and some neuroanatomy to characterize several classes of central interneurons that have different roles in state-dependent water-seeking behavior; these include the Janu-AstA neurons, which seem to function as a multi-modal integrator required for water seeking (but not water intake). The topic is certainly interesting and there are several novel findings. However, the paper lacks any accompanying neurophysiological analyses or careful control that the neuronal/molecular manipulations have the intended/expected effects, which limits the strength of the conclusions. Moreover, the sheer quantity and complexity of the work makes extraction of a compelling central discovery of broad general interest difficult to appreciate at this stage.

1. The manuscript is very difficult to follow: there is a large number of different types of behavioral assays (with flies in different internal states), and many different driver lines are presented as the authors home in on those that form the central novel findings (i.e., Janu neurons). The huge amount of data masks the main messages of the paper, and we'd strongly suggest the authors prune irrelevant experiments. For example, it jumps from characterization of the new hygrotactic assay (though not the first of this type of assay – – see minor comment below) to a screen using a completely different assay, which gives the impression the authors wanted to show the results of the screen rather than present a logical flow of experiments for the reader. In addition, inconsistent use of nomenclature in text/figures/legends renders it difficult to follow, e.g., Janu neurons are sometimes to referred by name, but often in the behavior data panels only the codes for the split-Gal4s are shown. Some summary schematics could help significantly, particular accompanying the Discussion sectionDiscussion section.

2. The authors infer roles for specific neurons and neural signaling molecules using a variety of genetic tools, but there is no molecular, cellular or physiological level validation that links these manipulations to changes in neuronal function. This is important, because it remains unclear whether the phenotypic differences observed upon activation/inhibition of a population of neurons is due to the difference in their individual roles or to the difference in the magnitude of activation/inhibition. (One might be activating a lot of neurons that have the same role in water seeking and thus the phenotype is stronger than activating only a subpopulation of those neurons). Many RNAi experiments are presented but, while several do point towards a consistent conclusion, there is no evidence that particular knock-downs work or do not work, and single mutant alleles are shown without rescue or obvious control of genetic background (for both Npf and AstA mutants, a w strain is used as the control).

3. No physiological experiments are presented (calcium imaging or electrophysiology), making it difficult to appreciate the function of individual neurons. For example, a simple question is whether Janu-AstA and Janu-GABA neurons respond to water vapor and/or water contact? We appreciate such experiments are probably beyond the scope of a reasonable revision but are part of the expected standard in the field now to make claims that the "neural basis" of a particular process is understood.

*Reviewer #3:*

Landayan, Zhou, and Wolf have identified a population of neurons that affect the relative drive strengths of thirst and hunger, and thus the choice between consummatory behaviors. Subsets of inhibitory and Allatostatin A-positive neurons seem to promote thirst.

Previous work on the interplay between hunger and thirst has focused on sensory circuits, so implicating central neurons, as proposed here, would be a significant contribution to understanding this process. Going from an unbiased behavior screen to a subset of neurons docked into a circuit is admirable. The work is careful, thorough, and on an important topic, so overall, I support publication, but the presentation needs some revisions to make the most supportable findings clearer to the reader.

The data order in the text and the figures do not align well, making it difficult to extract the conclusions and the evidence for them. There is a lot of chronological and extraneous data included that could be moved to supplementary figures to streamline. In general, the data on thirst is stronger than that on NPF and feeding, so the thirst conclusions should be emphasized.

The paper would benefit from a few schematics: one showing the proposed circuit, and a another showing the nested/Venn diagram of Durstig, Janu, AllatostatinA+, and GABA+ neurons, (maybe including their respective GAL4 targeting reagents).

The authors should be commended for wrestling with a complex story and for being open about all of the data even when it is hard to reconcile with a simple model, but in the end the insights need to be distilled more for public consumption.

One suggestion is to describe the original identification of Durstig neurons as thirst-promoting, then how a subset within that was identified as critical and Allatostatin-A-positive, and then demonstrate that this subset promotes thirst as a drive, suppressing the hunger drive. With a serious organizational revision and streamlining of extra data into supplementary figures, the findings should be clearer and much easier to convey.

[Editors’ note: further revisions were suggested prior to acceptance, as described below.]

Thank you for resubmitting your work entitled "Thirst interneurons that promote water seeking and limit feeding behavior in *Drosophila*" for further consideration by *eLife*. Your revised article has been evaluated by a Senior Editor and a Reviewing Editor.

All of reviewers appreciated the quality of the data, the potential interest of the work as well as the amount of work undertaken by your group, but they also concurred with regret that the data remained confusing and that conclusions reached would need to be supported by significantly more experimental work, some of which would not likely not be feasible with current tools. Their individual comments are detailed below.

*Reviewer #1:*

I remain ambivalent about this manuscript. On the one hand, the authors have responded to most of our requests on the original version with additional experiments and reformatting. On the other, it is still a complex story to navigate, and is still not very conclusive. I sense the authors have gone as far as is practically possible in this project with the resources available. While they have identified a potentially very interesting set of neurons, thirst seeking is clearly an intricate behavior to dissect using neurogenetics. The work opens a number of interesting questions, but I feel the level of evidence here is lower than similar types of papers in this field here. In addition, because I don't think they can make extremely strong claims about the biological role of these neurons and the circuit in which they are embedded, the breadth of interest beyond the immediate field is still relatively limited.

The authors have identified multiple types of "Janu" neurons with specific anatomical and neurochemical properties (e.g., GABA, AstA). These neurons all control thirst seeking, and also other unique (but related) behaviors. The Janu-AstA neuron is argued to synapse with and signal to NPF neurons. Focusing in on the key data in support of these main claims, I am only mildly convinced. The strongest phenotypes they observe are always with the broad drivers (e.g. Durstig-Gal4, which has widespread expression through the brain), but once they home in on a split-Gal4 combination with very selective labeling, the phenotype effects of activation or suppression are very small (e.g. Figure 2J-K or the new optogenetic activation experiments in Figure 5E). For some experiments, they revert to using the broader driver, such as the GRASP with NPF neurons using R65D05-Gal4. Unfortunately, this Gal4 line is expressed in a mass of neurons in this region of the brain (Figure 2S1A), so it's not clear to me that a positive GRASP signal really reflects synaptic contact between Janu-AstA neurons (somewhere within the R65D05 pattern) and NPF neurons. As mentioned in the previous round of reviews, the baseline phenotypes of controls is variable (e.g. in Figure 3I and Figure 3S1A VGAT RNAi experiments, there are many fewer flies on H2O compared to those with Gad1 RNAi in Figure 3J). The authors somewhat dismissed this concern by arguing that this may reflect variable environmental conditions and water deprivation treatments of flies of experiments on different days (and they only compared assays run in parallel), but this remains a concern for me. The n of several assays seems rather low (6-7), so it's not clear if they did all experiments of one type on a single day. In any case, questions linger as to how robust some of their results are.

In response to the original request to examine the physiological response of Janu neurons to water deprivation, the authors have used the TRIC sensor, a measure of long-term changes in neuronal calcium. This is not yet an extremely widely-used reporter, and I suspect requires careful controls to be able to interpret properly. They observe that one of the Janu-GABA neurons, but not any of the other three Janu neurons, displays increased TRIC signal upon water deprivation. This is interesting, but I find it puzzling that it should so selective, given that all of these neurons are behaviorally implicated in thirst seeking. I'd be interested to know what TRIC signal would be observed with a broad driver (e.g. Durstig-Gal4); is the single Janu-GABA neuron the only one that would have a significant increase in TRIC fluorescence? Water deprivation presumably induces widespread physiological changes in the brain, some of which may be direct sensors, and others show indirect activity. Ultimately, I don't find the new data so compelling in revealing this neuron to be a special sensor of thirst. (I appreciate more acute measurements of physiological activity are beyond the scope of the current work).

Although the authors have streamlined the figures, the reader is still directed to the many ancillary experiments in supplementary figures, now often with little explanation (e.g. Figure 1S1A-B, are only obliquely referred to in the text as "Neuronal activation and inactivation experiments uncovered additional behavioral roles for neurons in the Durstig pattern", and I don't fully understand the results in these panels). There is still an awkward transition at line 99 from the behavioral characterization of thirst-seeking to the screening for neuron populations promoting "seeking behavior"; ultimately, I find the Durstig-Gal4 neuron data almost impossible to interpret given its widespread expression pattern. On line 127 they say that "subtracting R65D05 neurons from Durstig reduced activation-dependent water-seeking, indicating that the water seeking neurons in Durstig and R65D05 are the same"; this is true, but there is still substantial water-seeking behavior implying there are other seeking-promoting neurons in the Durstig pattern. Are these Janu-GABA neurons or others not characterized here?

As this is a review of a revised manuscript, and as I believe the authors have gone as far as they can in the current revision, I do not propose further experimentation, as I believe a decision should be made based solely on the data now available.

*Reviewer #2:*

In this paper, Landayan et al. characterize two pairs of *Drosophila* neurons (called Janu) involved in orientation behaviors toward water following water deprivation (called thirst here). One pair of these neurons is GABAergic (Janu-GABA), while the other is producing allatostatin A (Janu-AstA). The anatomy, projections and role in the behavior of these neurons is dissected in detail using a host of approaches and genetic intersectional approaches. By manipulating the activity of either of these neurons, the authors show that Janu neurons motivate flies to seek humidity but that they are not promoting drinking. They also confirm earlier studies showing that motivation to search for humidity are antagonist to food searching. These results are new, interesting, and also puzzling because they indicate that the activation of only a few interneurons has a profound influence on complex behaviors, the authors suggesting that these neurons are linked to motivational states (like thirst).

While the experiments and the results reported here seem sound, the paper is difficult to read for those unfamiliar to the field. The starting point of this study is the use of a simple behavioral test to evaluate the aggregation of flies next to food or water to screen a library of Gal enhancer trap flies (InSite), while activating the corresponding cells (using TrpA1 and heat). They find one enhancer-GAL4 mutant called Durstig (thirst in German), which is particularly attracted by food and by humidity when the corresponding cells are activated. From the description of this behavior (ms lines 99-110), it is not clear if Durstig neurons are only involved in the attraction toward humidity or also toward food odors.

In a first step of their research, the authors further used this behavioral test to evaluate if other genetic constructions could reproduce the behavioral pattern observed in Durstig> TrpA1 flies. Only two of these constructions reproduced the water-seeking pattern of Durstig flies. One is described as the "NP883 pattern", but this pattern has not been further compared to Durstig neurons. The second is called "R65D05 pattern" which was shown here as including Durstig neurons responsible for humidity searching.

In order to narrow down the Durstig neurons responsible for thirst-related attraction behavior, they used Gal80 constructions to subtract out neurons with various neurotransmitters. Finally, the authors were able to narrow down the neurons responsible to the attraction to 2 pairs of neurons characterized / defined by a split-Gal4 construction called Janu (again meaning thirst but this time in Estonian). According to figure 2G, this construction combines R52A01-AD and D65D05-DBD. It is thus not directly related to Durstig, except that activating the Janu cells induces the same behavioral phenotype related to humidity attraction and thirst.

What is next described in the paper is a thorough analysis of the Janu neurons, ie their anatomical location (in the adults), and the finding that two are GABAergic, while the two others are expressing allatostatin A. They also characterize their function and role in the behaviors studied here.

As compared to the first version, the ms is much improved and easier to understand. Besides the factual results shown here, it represents a very interesting work illustrating top-notch genetic strategies to identify neurons responsible for specific behaviors.

A general sketch of the Janu neurons, summarizing their role in thirst, and their relation to feeding, etc, might be useful.

*Reviewer #3:*

The authors use an unbiased screen for neurons that contribute to feeding behavior to identify new circuits that affect thirst. Some of these neurons are GABAergic, while others release the neuropeptide Allatostatin, and they connect to the NPF-positive neurons previously implicated in hunger sensation. The interplay between thirst and hunger is complex and a better understanding of how these neurons interact with sensory and motor components will help illuminate how flies balance these conflicting motivational drives.

The authors have clearly made a good faith effort to address the issues raised in the initial review of this manuscript.

There is an enormous amount of work presented, and there is the potential that it reveals neurons that contribute to appropriate behavioral responses to hunger and thirst. Understanding how neural circuits function to regulate choices between essential and competing drives is a really important area of research.

But I remain very confused. There is still no summary diagram for either the (known and new) neuronal circuitry or the interaction between the motivational drives. The brief flow diagram of the response to thirst helps a bit but does not really clarify. Some neurons seem to be able to induce a motivational state (or the behavioral response associated with it) while others are partially but not completely required for an appropriate choice between eating and drinking? I think the Janu-GABA and Janu-allatostatin neurons are different populations, but I am actually not sure.

The figures are very dense and intermix feeding and drinking behaviors, activation and inactivation experiments, and different sets of neurons. The combination thirst response and place preference in figure 5, for example, makes no sense at all.

The anti-VGAT staining seems to label the entire neuropil, which makes the co-localization with some of the Janu neurons very difficult to interpret. The allatostatin colocalization is similarly confusing.

This manuscript still needs a major overhaul to make its contributions to thirst circuitry interpretable.

---

## [Author Response]

[Editors’ note: the authors resubmitted a revised version of the paper for consideration. What follows is the authors’ response to the first round of review.]

Essential revisions:1. (A) A major rewrite is required. While the authors may be commended for wrestling with a complex story and for being open about all of the data even when it is hard to reconcile with a simple model, in the end the insights need to be distilled more for public consumption. In current form, the paper is extraordinarily difficult to read and difficult to follow, in part due to the complexity of the methods. There is a large number of different types of behavioral assays (with flies in different internal states), and many different driver lines are presented as the authors home in on those that form the central novel findings (i.e., Janu neurons). (B) The huge amount of data masks the main messages of the paper. There is a lot of chronological and extraneous data included that could be moved to supplementary figures to streamline and we'd strongly suggest the authors prune irrelevant experiments. For example, it jumps from characterization of the new hygrotactic assay (though not the first of this type of assay – – see minor comment below) to a screen using a completely different assay, which gives the impression the authors wanted to show the results of the screen rather than present a logical flow of experiments for the reader. In addition, inconsistent use of nomenclature in text/figures/legends renders it difficult to follow, e.g., Janu neurons are sometimes to referred by name, but often in the behavior data panels only the codes for the split-Gal4s are shown. (C) In general, the data on thirst is stronger than that on NPF and feeding, so the thirst conclusions should be emphasized. One suggestion is to describe the original identification of Durstig neurons as thirst-promoting, then how a subset within that was identified as critical and Allatostatin-A-positive, and then demonstrate that this subset promotes thirst as a drive, suppressing the hunger drive. With a serious organizational revision and streamlining of extra data into supplementary figures, the findings should be clearer and much easier to convey. Some summary schematics could help significantly, particular accompanying the Discussion section. For example: one showing the proposed circuit, and a another showing the nested/Venn diagram of Durstig, Janu, AllatostatinA+, and GABA+ neurons, (maybe including their respective GAL4 targeting reagents).

A. The manuscript is completely rewritten and completely reorganized. In addition, extraneous data is removed and new data is added as detailed below. Briefly, we de-emphasize the process of isolating the individual thirst water seeking neurons to a single figure, and we now focus on their characterization. Moreover, we completed additional experiments to make each thirst neuron characterization follow the same logical path and to be more complete. The main story presented in the original manuscript remains valid, however new data allows us to make stronger and clearer conclusions about the role of each interneuron in thirst and hunger.

B. We removed extraneous experiments, moved many others into supplemental figures, and added explanatory diagrams to help guide the reader. We think that the main figures focus clearly on thirsty water seeking. Moreover, the emphasis is not on finding the neurons but instead their characterization.

C. We emphasize the conclusions on Janu-GABA, Janu-AstA, and the relation of NPF to Janu-AstA. Our findings for NPF are now clearer with the addition of new data.

2. (A) One problem than runs through the manuscript is that it remains formally unclear whether the phenotypic differences observed upon activation/inhibition of populations of neurons is due to the difference in their individual roles or to the difference in the magnitude of activation/inhibition. (One might be activating a lot of neurons that have the same role in water seeking and thus the phenotype is stronger than activating only a subpopulation of those neurons). (B) Similarly, many RNAi experiments are presented but, while several do point towards a consistent conclusion, there is no evidence that particular knock-downs work or do not work, (C) and single mutant alleles are shown without rescue or obvious control of genetic background (for both Npf and AstA mutants, a w strain is used as the control). Some of these issues need to be addressed experimentally and certainly in the text.

A. This issue is largely resolved by focusing our characterization on individual neurons (Janu-GABA, Janu-AstA, and AstA receptor expressing NPF neurons). It is also resolved by focusing on water seeking almost exclusively, and only testing for feeding behavior at strategic points.

B. RNAi experiments were done with previously characterized RNAi transgenes (for example, AstA receptor, but true for almost every RNAi transgene used in this study, as documented in the Results and in Table S2), with distinct RNAi transgenes targeting the same gene (for example AstA), and targeting different genes that are expected to affect the process in the same way (for example Gad1 and VGAT to decrease GABA synthesis).

C. In new data, we placed the mutants (all previously characterized in the literature) in trans to small deficiencies and validated the loss-of-function behavioral effects. The mutant alleles were outcrossed to the genetic background strain prior to behavioral testing.

3. Many of the issues above could be addressed by physiological experiments (calcium imaging or electrophysiology), which would potentially make it easier to appreciate the function of individual neurons. For example, a simple question is whether Janu-AstA and Janu-GABA neurons respond to water vapor and/or water contact? We appreciate such experiments are probably beyond the scope of a reasonable revision. But in the absence of this, extra care should be paid to both establishing the specificity of the genetic perturbations, the resulting phenotypes and in interpretation of the results. As the quantity and complexity of the work makes extraction of a compelling central discovery of broad general interest difficult to appreciate, major revisions should concentrate on marshalling and presenting all the best evidence to arguing for and documenting the identification of thirst-promoting central neurons.

We used TRIC to demonstrate that both the Janu-GABA neuron and NPF neurons are activated by thirst. Neither Janu-GABA nor Janu-AstA are activated by hunger. NPF neurons are previously shown to be activated by hunger. Nor did TRIC identify any effect of water vapor or water intake on Janu neuron activity in thirsty flies. However, it is possible that TRIC may not have the appropriate sensitivity to detect these responses, and therefore we excluded this negative data. We also collaborated with the lab of Tom Clandinin at Stanford to test for GCaMP6 responses in Janu neurons with optogenetic activation of Ir68a humidity sensing neurons, but we were unable to detect any signal in either water replete or thirsty flies.

We wish to emphasize the current complexity of measuring thirst-related neuronal activity in flies, due to the effects of hemolymph osmolarity on thirst circuitry, as shown by Kristin Scott’s lab at UC Berkeley: The dissections needed for imaging impact interstitial fluid composition and so thirst neuron activity. The level of expertise needed to perform these experiments is quite high, and it is beyond the means of my lab (at an R2 university that is only 15 years old, three neurobiology labs total at the university, a graduate student-driven lab, and this project is a new direction for my lab that was funded off of my start-up funds). Through collaboration we will be able to conclusively address neuronal activity using in vivoin vivo real time activity sensors with 3-photon microscopy or similar technology, where imaging can be done with intact animals. However, at the moment this is out of our reach.

Finally, we had an optogenetic device built expressly to address if flies prefer activation of Janu neurons, and we now report that they do. This adds a new functional dimension to our findings and provides a more accurate description of Janu neuron function.

4. The title and the summary are misleading, as the Janu neurons described here are not only allatostatin producing but also GABAergic. Perturbations specific to Allostatin/Janu neurons form only a relatively small part of the entire analysis, which ignores that other Janu neurons are GABAergic. This should be explicitly clarified. Similarly, in the abstract the phrase: "allowed us to uncover the neural basis.……" is a bit of an over-statement (perhaps "a neural basis").

The title is updated to reflect our main findings. The abstract, like the rest of the manuscript, is completely rewritten.

Author response table 1 describes the most consequential of the new experiments in the manuscript.

**Author response table 1. resptable1:** 

Figure	Question	Approach	Result	Interpretation
3K	Are Janu-GABA neurons important for neural activation driven seeking?	We co-expressed a GABA neurotransmitter RNAi with TrpA1 activation.	Water seeking is suppressed.	Janu-GABA neuron activation drives replete water seeking.
3L,M 3S1B,C	Are Janu-GABA neurons important for water ingestion?	We used well-established behavioral assays to specifically address ingestive behaviors.	Water ingestion does not change. But, the first bout of water intake decreases. Pumping rate is unchaged.	The first bout of water intake is mediated by Janu-GABA neurons. Other downstream neurons may be responsible for the maintenance of persistent ingestive behaviors.
3N	Are Janu-GABA neurons important for fed and hungry feeding behavior?	We expressed a GABA neurotransmitter RNAi and assessed dry sucrose occupancy in fed and hungry flies.	Fed and hungry dry sucrose occupancy is unaltered.	Janu-GABA neurons are specific for thirst and do not contribute to feeding behavior.
4L	Are Janu-AstA neurons important for neural activation driven seeking?	We co-expressed an AstA neurotransmitter RNAi with TrpA1 activation.	Water seeking is suppressed	Janu-AstA neuron activation drives replete water seeking.
4M,N	Are Janu-AstA neurons important for water ingestion?	We used well-established behavioral assays to specifically address ingestive behaviors.	Water ingestion and first bout of water intake does not change. Pumping rate is unchanged.	Janu-AstA neurons are dispensable for thirsty water ingestion.
5A,B,C	Are Janu-GABA and Janu-AstA activated by states of thirst and hunger?	We used TRIC, a calciumdependent fluorescent reporter, to assess neuronal activity.	Only the Janu-GABA1 subtype was highly active in thirsty flies. Janu-AstA was not active in thirsty or hungry states.	The Janu-GABA1 neuron is thirst activated. It may be regulated by the osmosensory ISNs.
5D,E 5S1A,B,C	Does the activity of Janu and R65D05 neurons encode positive or negative valence?	We optogenetically activated Janu and R65D05 neurons to assess positional preference.	Flies exhibit positional preference for activation of Janu or R65D05 neurons.	Janu and R65D05 neurons encode positive valence. This is distinct from mammalian hunger and thirst neurons which encode negative valence.
6EF 6S1B	Is AstA-R2 signaling on NPF neurons important for thirsty water ingestion?	We expressed an AstA-R2 RNAi in NPF neurons. We used well-established behavioral assays to specifically address ingestive behaviors.	Total water consumption and first bout water consumption are decreased. Pumping rate is unaffected	AstA signaling onto NPF neurons is important for water intake.
6H	Is NPF signaling important for thirsty water seeking?	We expressed a NPF RNAi in thirsty flies to assess water seeking.	Water seeking is increased.	NPF released by NPF neurons suppresses water seeking. Thus, NPF can reciprocally modulate hunger and thirst.
6I 6S1C	Are NPF neurons activated by thirst?	We used TRIC, a calcium-dependent fluorescent reporter, to assess neuronal activity.	Dorsomedial P1 and lateral L1 NPF neurons both are significantly activated in thirsty states.	NPF neurons encode thirst and are a critical convergence node for the integration of thirst and hunger signals.

[Editors’ note: what follows is the authors’ response to the second round of review.]

Reviewer #1:I remain ambivalent about this manuscript. On the one hand, the authors have responded to most of our requests on the original version with additional experiments and reformatting. On the other, it is still a complex story to navigate, and is still not very conclusive. I sense the authors have gone as far as is practically possible in this project with the resources available. While they have identified a potentially very interesting set of neurons, thirst seeking is clearly an intricate behavior to dissect using neurogenetics.

A comprehensive description of the thirst circuitry is an impossible task for any lab, given its multiphasic character and its complex relation to other brain functions. Moreover, there is hardly anything known about the central circuitry for thirst; our work identifies components of it. We went from zero to two well-described neurons, and two neurons that have not had any function ascribed to them previously, using unbiased techniques.

Neurogenetics is a remarkable tool for dissecting thirst behavior – our work and that of many other labs is evidence of that. Plus, we extensively used behavior, anatomy, functional characterization, and some physiological tests of neuronal activity.

The work opens a number of interesting questions, but I feel the level of evidence here is lower than similar types of papers in this field here.

Two examples of similar types of papers:

1. Perhaps the best example is characterization of ITP as an endocrine regulator of water homeostasis from the Nässel lab in PLoS Genetics (https://doi.org/10.1371/journal.pgen.1007618). Here, the authors characterized the role of ITP in a variety of behaviors related to thirst and hunger. How they differ from us is they started with a peptide previously implicated in ionic balance, and the majority of the manuscript uses a pan-organismal driver to knock down ITP and assess phenotypes. Thus, we do not yet know if ITP functions in the same or different cells (or which cells, neuronal or otherwise) for water vs. food ingestion. By contrast, we de novo identified individual neurons that regulate specific steps of thirst representation through specific transmitters and modulators. We show functional connectivity between the Janu-AstA and NPF neurons for reciprocal regulation of water seeking and feeding behavior.

2. A second example is the regulation of thirst and hunger through the ISN neurons from the Scott lab in Cell (http://dx.doi.org/10.1016/j.cell.2016.06.046). Here the authors performed a neuroanatomical screen to identify the ISN neurons that they show promote food intake and inhibit water intake in an osmosensitive manner. This paper is an exceptionally well-done characterization of a single set of neurons, yet they describe no circuit connectivity, no neurotransmitter/neuromodulator for the ISNs, and test a very narrow range of behavioral characterization (food and water intake). Thus, while it is difficult to find fault with the research presented, it is also very limited in the scope for understanding their function.

At its core, our work is an example of how classic neurogenetic manipulations can provide a solid foothold for the scientific community, here to understand how thirst is encoded in the central brain, and its interactions with hunger.

To summarize, we showed:

1. that AstA in Janu-AstA is necessary and sufficient for water seeking, that it is specific for the seeking step of the thirst behavioral sequence, that it reciprocally suppresses feeding behavior, that it synapses onto NPF neurons, that the AstA-R2 receptor functions in NPF neurons for thirsty seeking and reciprocally feeding behavior, that NPF in NPF neurons regulates thirsty seeking, and that thirst increases the activity of specific NPF neurons.

2. that Janu-GABA neurons are activated by thirst, and that GABA in Janu-GABA is highly specific to thirsty water seeking.

We note that the neuromodulators AstA and NPF were previously known to regulate feeding behaviors but not thirst.

We make a guess that the reviewer is noting the lack of state-of-the-art electrophysiology and in vivo GCaMP imaging in awake behaving animals. However, we believe thirst circuitry physiology, in particular, awaits better establishment of non-invasive imaging techniques like 3-photon, due to the unique need to preserve physiological osmolarity.

In addition, because I don't think they can make extremely strong claims about the biological role of these neurons and the circuit in which they are embedded, the breadth of interest beyond the immediate field is still relatively limited.

This work is foundational: we identify new neurons for thirsty seeking of water. This is in contrast to providing new characterization of previously described neurons that is more common in the literature of the circuitry of behavior (for example, the mushroom body circuitry or the P1 neurons in courtship and aggression). Indeed, even those well-characterized circuits still give up new information for their function, so completeness is relative (see also limits of ITP and ISN characterization in the previous comment). Our work provides a clear basis for new studies on thirst and its relationships to other internal states, and the specific behavioral steps that compose it.

Most surprisingly, the Janu neurons we characterize can simultaneously antagonize competing states like hunger through interactions with NPF neurons. In the mammalian literature, thirst can suppress feeding behaviors. The neural mechanism is executed by glutamatergic neurons in the subfornical organ and are critical for this thirst-dependent hunger suppression (Zimmerman et al., Nature. 2017, Figure S7 CD), however the hierarchical wiring between thirst and hunger neurons is only beginning to be constructed and is an active area of research (Gong et al., Cell. 2020). We hope that our work will shed light on the fundamental characteristics of one type of thirst neuron (Janu neurons) and provide an entry point into the discovery of even more thirst neurons that may work in a parallel or redundant fashion, in concordance with homeostatic neuron functionality (Betley et al., Cell. 2013).

The authors have identified multiple types of "Janu" neurons with specific anatomical and neurochemical properties (e.g., GABA, AstA). These neurons all control thirst seeking, and also other unique (but related) behaviors. The Janu-AstA neuron is argued to synapse with and signal to NPF neurons. Focusing in on the key data in support of these main claims, I am only mildly convinced. The strongest phenotypes they observe are always with the broad drivers (e.g. Durstig-Gal4, which has widespread expression through the brain), but once they home in on a split-Gal4 combination with very selective labeling, the phenotype effects of activation or suppression are very small (e.g. Figure 2J-K or the new optogenetic activation experiments in Figure 5E).

Specific reduction of either GABA in Janu-GABA (Figure 3J) or AstA in Janu-AstA (Figure 4H) strongly inhibits thirsty seeking, indicating their critical role in the behavior. Thus the critique of “strongest behavior in the broadest driver” is limited to the activation experiments. Why the reduced activation (TrpA1) and inactivation (Shi) effects in 2J-K may be due to weaker driving of these dose dependent heat activated tools), however we note that the results are significant. The Janu-GABA and JanuAstA neurons are clearly critical for thirsty seeking.

Milder behavioral phenotypes are commonly observed with split-Gal4s. For example, activation of single glutamatergic mushroom body output neurons (MBONs) that tile the horizontal lobe will induce avoidance. However, manipulation of multiple groups of glutamatergic MBONs results in a stronger and synergistic avoidance phenotype (Aso et al., *eLife*. 2014. Figure 4). It is possible that JanuAstA is one type of AstA neuron (or non-AstA neurons in R65D05) that can promote water seeking. We also note that Janu is a weaker Gal4 driver for the Janu-AstAs than R65D05, assessed by myristoylated-GFP expression levels, so it may simply be Gal4 driver strength.

For some experiments, they revert to using the broader driver, such as the GRASP with NPF neurons using R65D05-Gal4. Unfortunately, this Gal4 line is expressed in a mass of neurons in this region of the brain (Figure 2S1A), so it's not clear to me that a positive GRASP signal really reflects synaptic contact between Janu-AstA neurons (somewhere within the R65D05 pattern) and NPF neurons.

GRASP between Janu and NPF was technically impossible because NPF-LexA and R52A01-AD reside in the same attP40 landing site, and the GRASP halves are located nearby in the genome on the second chromosome. We tried and failed in making triple transgenic recombinants with GRASP to make Janu/NPF GRASP possible.

We agree that positive GRASP signals between Janu-AstA and NPF neurons would be explicitly clear, however we have provided orthogonal lines of evidence that argue Janu-AstA is pre-synaptic to NPF neurons:

A. These are AstA-positive directional GRASP signals between R65D05 (presynaptic) and NPF (postsynaptic) neurons. While R65D05 does contain other AstA-positive neurons in the SMP, the location at the extreme medial aspect of the SMP is Janu-AstA exclusively. We can provide immunohistochemical evidence.

B. AstA in Janu-AstA neurons promotes thirsty seeking (Figure 4H).

C. AstA-R2 is required in NPF neurons for thirsty seeking (Figure 6D).

As mentioned in the previous round of reviews, the baseline phenotypes of controls is variable (e.g. in Figure 3I and Figure 3S1A VGAT RNAi experiments, there are many fewer flies on H2O compared to those with Gad1 RNAi in Figure 3J). The authors somewhat dismissed this concern by arguing that this may reflect variable environmental conditions and water deprivation treatments of flies of experiments on different days (and they only compared assays run in parallel), but this remains a concern for me.

We argue that our approach may be better, albeit more variable, than those in prior publications studying thirst in *Drosophila*. Thirst is a linearly graded state that is read out by the intensity of seeking (Figure 1D, see also Pool et al. for intake; https://doi.org/10.1016/j.neuron.2014.05.006), so the absolute amount of water deprivation can vary in a range without compromising the results. Other labs that studied thirst typically take the flies to a fraction of lethality to ensure similar deprivation between experiments. We chose a more naturalistic and intermediate deprivation because it is likely that flies invoke emergency pathways when nearing death, and we specifically wanted to study thirst.

Moreover, we provide genetic controls for all experiments to distinguish differences in water seeking within an experiment.

Finally, biological variability between experimental controls is not uncommon in these types of behavioral assays. For example, in a similar ‘seeking assay’ using 24 hour food deprived flies, ‘control food seeking’ can ranges from 40-80% (Tsao et al., *eLife*. 2018. Figure 2,3,4,6, 7, 8, 10, 11)

The n of several assays seems rather low (6-7), so it's not clear if they did all experiments of one type on a single day.

We will amend the Methods to reflect that all experiments were done across multiple days. This approach to animal behavior was instilled into the senior author (Wolf) through training in the behavioral mechanisms of alcohol inebriation in flies, where day-to-day variability is much greater and is routinely factored into the experiments.

In any case, questions linger as to how robust some of their results are.In response to the original request to examine the physiological response of Janu neurons to water deprivation, the authors have used the TRIC sensor, a measure of long-term changes in neuronal calcium. This is not yet an extremely widely-used reporter, and I suspect requires careful controls to be able to interpret properly.

It is well-used and well-vetted in the literature, including but not limited to:

www.pnas.org/cgi/doi/10.1073/pnas.1706608114

https://doi.org/10.1038/s41593-019-0515-z

https://doi.org/10.1016/j.neuron.2018.07.001

https://doi.org/10.7554/eLife.54229

https://doi.org/10.1016/j.neuron.2019.04.009

http://dx.doi.org/10.7554/eLife.22441

Our use to measure internal state-dependent neural activity (longer time frame neuromodulatory events) is consistent with its prior uses.

We showed that the Janu-GABAs respond to thirst but not hunger, indicating that the reporter is not responding in some non-specific manner. An additional example, the NPF P1 neurons respond to both thirst and hunger, indicating TRIC can detect modulatory activity changes for distinct states in the same neuron.

They observe that one of the Janu-GABA neurons, but not any of the other three Janu neurons, displays increased TRIC signal upon water deprivation. This is interesting, but I find it puzzling that it should so selective, given that all of these neurons are behaviorally implicated in thirst seeking.

Yes, biology is richly complicated (thank goodness), and we captured aspects of that. We propose in the discussion that the Janu-AstA may be regulated by thirst below the detection limit of the tool. Alternatively, the regulation may be at the AstA-R2 receptors, with Janu-AstA providing a tonic signal. The latter mechanism is mechanistically analogous to sNPFR regulation by starvation to tune sensory input (https://doi.org/10.7554/*eLife*.08298).

I'd be interested to know what TRIC signal would be observed with a broad driver (e.g. Durstig-Gal4); is the single Janu-GABA neuron the only one that would have a significant increase in TRIC fluorescence? Water deprivation presumably induces widespread physiological changes in the brain, some of which may be direct sensors, and others show indirect activity. Ultimately, I don't find the new data so compelling in revealing this neuron to be a special sensor of thirst. (I appreciate more acute measurements of physiological activity are beyond the scope of the current work).

We propose that Janu-AstA promotes thirsty seeking of water and suppresses feeding behavior, and therefore is a critical neuron in the representation of thirst in the brain. That we can subtract nearly all water seeking from Durstig with R52A01 and R65D05, that we can subtract nearly all thirsty seeking from Janu with either Gad1 RNAi (Figure 3J) or AstA RNAi (Figure 4I) speaks to the importance of these neurons for thirst representation. AstA RNAi also strongly suppresses thirsty seeking from Durstig and R65D05 (Figure 4S1 DandE). The specificity for the seeking aspect of thirst strengthens the argument that it is not due to the creation of a competing drive (i.e. it is a thirst neuron).

The question of how many neurons change activity in thirst is fascinating: we predict widespread circuit activity, due to the multiphasic nature of thirst behavioral responses, and the interactions of thirst circuitry with other brain systems, as was recently demonstrated in mice (http://science.sciencemag.org/content/364/6437/eaav3932). Building tools to functionally capture and then manipulate these patterns will be crucial to future complete understanding of the parsing of internal and external sensory cues to select and carry out the appropriate behavior.

Our narrower scope approach is typical for behavioral assignment advances, while the scientific field designs functional approaches to determine the causal role of all the neurons (and glia) regulated by a state.

Although the authors have streamlined the figures, the reader is still directed to the many ancillary experiments in supplementary figures, now often with little explanation (e.g. Figure 1S1A-B, are only obliquely referred to in the text as "Neuronal activation and inactivation experiments uncovered additional behavioral roles for neurons in the Durstig pattern", and I don't fully understand the results in these panels).

This was an apparently bad decision by the senior author (Wolf) to leave in some of the earlier but confounding data from Durstig (1S1A-B). The data shows that, in a more complex expression pattern, additional neurons influence the measured behaviors; this is useful for future characterization of additional neurons, but we agree that it can be confusing for the readers. We will further streamline the presentation if allowed to do so, specifically eliminating 1S1A-B, and reassessing the value of others in the supplementals.

There is still an awkward transition at line 99 from the behavioral characterization of thirst-seeking to the screening for neuron populations promoting "seeking behavior"; ultimately, I find the Durstig-Gal4 neuron data almost impossible to interpret given its widespread expression pattern.

We agree that the use of “seeking behavior” to describe the TrpA1 activation screen is unclear.

The Durstig data should have been limited to our discovery that thirst neurons exist in the pattern. We will further streamline the presentation if allowed to do so, as noted in the previous reply. We intended for the reader to understand that we began the discovery of the thirst neurons with Durstig.

On line 127 they say that "subtracting R65D05 neurons from Durstig reduced activation-dependent water-seeking, indicating that the water seeking neurons in Durstig and R65D05 are the same"; this is true, but there is still substantial water-seeking behavior implying there are other seeking-promoting neurons in the Durstig pattern. Are these Janu-GABA neurons or others not characterized here?

The partial effects in Figure 2A (line 127) are likely due to the generally weaker activity of LexA drivers, here used to drive Gal80 to subtract R65D05 from Durstig.

Both the Janu-GABA and the Janu-AstA neurons are contained in Durstig, R65D05, and R52A01. Subtracting either R52A01 or R65D05 from Durstig decreases Durstig water seeking (Figure 2A, 2D), and thirsty seeking is readily blocked with inactivating the Janu split (Figures3J and 4H). We can add statements to ensure clarity. We included the subtraction experiments (and others) because they demonstrate that we isolated the Janu neurons from the Durstig pattern, and therefore are drivers of the original water seeking phenotype. There may be additional thirst neurons in the Durstig pattern. We demonstrated that neurons previously found to influence humidity preference (R48B04, likely a specific subset of PAM dopamine neurons) were not in Durstig.

As this is a review of a revised manuscript, and as I believe the authors have gone as far as they can in the current revision, I do not propose further experimentation, as I believe a decision should be made based solely on the data now available.

Thank you for your comments.

Reviewer #2:In this paper, Landayan et al. characterize two pairs of Drosophila neurons (called Janu) involved in orientation behaviors toward water following water deprivation (called thirst here). One pair of these neurons is GABAergic (Janu-GABA), while the other is producing allatostatin A (Janu-AstA). The anatomy, projections and role in the behavior of these neurons is dissected in detail using a host of approaches and genetic intersectional approaches. By manipulating the activity of either of these neurons, the authors show that Janu neurons motivate flies to seek humidity but that they are not promoting drinking. They also confirm earlier studies showing that motivation to search for humidity are antagonist to food searching. These results are new, interesting, and also puzzling because they indicate that the activation of only a few interneurons has a profound influence on complex behaviors, the authors suggesting that these neurons are linked to motivational states (like thirst).

Thank you.

While the experiments and the results reported here seem sound, the paper is difficult to read for those unfamiliar to the field. The starting point of this study is the use of a simple behavioral test to evaluate the aggregation of flies next to food or water to screen a library of Gal enhancer trap flies (InSite), while activating the corresponding cells (using TrpA1 and heat). They find one enhancer-GAL4 mutant called Durstig (thirst in German), which is particularly attracted by food and by humidity when the corresponding cells are activated. From the description of this behavior (ms lines 99-110), it is not clear if Durstig neurons are only involved in the attraction toward humidity or also toward food odors.

We will clarify by focusing on Durstig as a starting point to find the thirst interneurons, dropping our additional characterization of the pattern. The Durstig pattern likely contains multiple neurons that influence a variety of behaviors, including but not limited to thirst and feeding behaviors.

In a first step of their research, the authors further used this behavioral test to evaluate if other genetic constructions could reproduce the behavioral pattern observed in Durstig> TrpA1 flies. Only two of these constructions reproduced the water-seeking pattern of Durstig flies. One is described as the "NP883 pattern", but this pattern has not been further compared to Durstig neurons. The second is called "R65D05 pattern" which was shown here as including Durstig neurons responsible for humidity searching.In order to narrow down the Durstig neurons responsible for thirst-related attraction behavior, they used Gal80 constructions to subtract out neurons with various neurotransmitters. Finally, the authors were able to narrow down the neurons responsible to the attraction to 2 pairs of neurons characterized / defined by a split-Gal4 construction called Janu (again meaning thirst but this time in Estonian). According to figure 2G, this construction combines R52A01-AD and D65D05-DBD. It is thus not directly related to Durstig, except that activating the Janu cells induces the same behavioral phenotype related to humidity attraction and thirst.

Subtracting either R52A01 or R65D05 from Durstig blocks Durstig water seeking (Figure 2A, 2D), thus the Janu neurons are in the Durstig pattern.

What is next described in the paper is a thorough analysis of the Janu neurons, ie their anatomical location (in the adults), and the finding that two are GABAergic, while the two others are expressing allatostatin A. They also characterize their function and role in the behaviors studied here.As compared to the first version, the ms is much improved and easier to understand. Besides the factual results shown here, it represents a very interesting work illustrating top-notch genetic strategies to identify neurons responsible for specific behaviors.

Thank you.

A general sketch of the Janu neurons, summarizing their role in thirst, and their relation to feeding, etc, might be useful.

Summary of thirst circuitry. The ISN senses hemolymph osmolarity to switch between thirst and hunger, likely setting in motion the full thirst behavioral program when the ISN is turned off by high osmolarity (Jourjine et al., 2016). Janu-AstA and Janu-GABA are independently necessary and sufficient for promoting the water seeking step of thirst. Janu-AstA also inhibits feeding behavior, releasing AstA that acts through AstA-R2-expressing NPF neurons to regulate NPF release. AstAR2 expressing NPF neurons also regulate water intake independently of the Janu-AStA neurons.

Reviewer #3:The authors use an unbiased screen for neurons that contribute to feeding behavior to identify new circuits that affect thirst. Some of these neurons are GABAergic, while others release the neuropeptide Allatostatin, and they connect to the NPF-positive neurons previously implicated in hunger sensation. The interplay between thirst and hunger is complex and a better understanding of how these neurons interact with sensory and motor components will help illuminate how flies balance these conflicting motivational drives.The authors have clearly made a good faith effort to address the issues raised in the initial review of this manuscript.

Thank you.

There is an enormous amount of work presented, and there is the potential that it reveals neurons that contribute to appropriate behavioral responses to hunger and thirst. Understanding how neural circuits function to regulate choices between essential and competing drives is a really important area of research.But I remain very confused. There is still no summary diagram for either the (known and new) neuronal circuitry or the interaction between the motivational drives.

Please see the summary diagram which is included as Figure 6.

The brief flow diagram of the response to thirst helps a bit but does not really clarify. Some neurons seem to be able to induce a motivational state (or the behavioral response associated with it) while others are partially but not completely required for an appropriate choice between eating and drinking?

The Janu-GABA neurons are activated by thirst, and they specifically encode thirsty water seeking. We did not detect Janu-AstA activation by thirst, however they, like the Janu-GABAs, are required for thirsty seeking. The Janu-AstA neurons also reciprocally inhibit feeding behavior. These behavioral characterizations are done in detail in Figures 3 and 4. The summary diagram for Figure 6 will encapsulate these findings.

I think the Janu-GABA and Janu-allatostatin neurons are different populations, but I am actually not sure.

Yes, they are. That is shown anatomically in the flip-out experiments (Figure 2S4), and anatomically and functionally in Figures 3 and 4.

The figures are very dense and intermix feeding and drinking behaviors, activation and inactivation experiments, and different sets of neurons. The combination thirst response and place preference in figure 5, for example, makes no sense at all.

Figure 5 is a marriage of necessity – Both experiments are important for understanding thirst neuron function, and this is the location they belong in the story flow, yet they do not neatly fit into the other figures. Also, both address the Janu group as a whole, which is necessary at this point, right after we defined the function of the Janu-GABA and Janu-AstA neurons (Figures 3 and 4), and right before we provide evidence that the Janu-AstA neurons connect to NPF neurons (Figure 6). We hope that this is a minor concern.

Finally, we will color code thirst vs feeding behaviors, to help readers distinguish the experiments.

The anti-VGAT staining seems to label the entire neuropil, which makes the co-localization with some of the Janu neurons very difficult to interpret. The allatostatin colocalization is similarly confusing.

GABAergic neurotransmission is widespread throughout the adult *Drosophila* brain. The anti-VGAT staining is specific: there was no overlap of anti-VGAT staining in the Janu-AstA neurons, and the anti-VGAT overlap with the Janu-GABAs was limited to the larger varicosities in the Janu-GABA neurons, which typically are presynaptic regions of the neurons. Moreover, nearly all of the JanuGABA varicosities were VGAT-positive, which would not be expected for non-specific staining. We used anti-VGAT according to the conditions described by the David Krantz lab at UCLA that produced and characterized the antibody. We can elaborate this in the manuscript to help clarify.

The Allatostatin localization is in the cell bodies of the Janu-AstA neurons (Figure 4B-D) and in their presynaptic release sites (Figure 4E-G), as defined with neuronal polarity markers in Figure 2I.

Neuromodulator staining in both the cell bodies and the endings is typical (for example extensively documented in the NPF literature). The Janu-AstA neurons are the only AstA-positive neurons in the Janu expression pattern (4S1). AstA knockdown in Janu neurons blocks thirsty seeking (Figure 4H). Like GABA, there are additional AstA neurons outside of Janu – we are specifically studying those in the Janu pattern.

This manuscript still needs a major overhaul to make its contributions to thirst circuitry interpretable.

We will do our best to decrease confusion by providing additional guiding diagrams as suggested, and by eliminating some of the supplemental material that is not central to the main conclusions (also as suggested). Some of the genetic techniques are moderately difficult to understand, and we will try to help the readers further with explanatory diagrams.